# Anomalous efficiency elevation of quantum-dot light-emitting diodes induced by operational degradation

Siyu He[1], Xiaoqi Tang[1], Yunzhou Deng ●[2,3] ✉, Ni Yin[4], Wangxiao Jin[1], Xiuyuan Lu[1], Desui Chen[1], Chenyang Wang[1], Tulai Sun[5], Qi Chen ●[4] ✉ & Yizheng Jin ●[1] ✉

Quantum-dot light-emitting diodes promise a new generation of high-performance and solution-processed electroluminescent light sources. Understanding the operational degradation mechanisms of quantum-dot light-emitting diodes is crucial for their practical applications. Here, we show that quantum-dot light-emitting diodes may exhibit an anomalous degradation pattern characterized by a continuous increase in electroluminescent efficiency upon electrical stressing, which deviates from the typical decrease in electroluminescent efficiency observed in other light-emitting diodes. Various in-situ/operando characterizations were performed to investigate the evolutions of charge dynamics during the efficiency elevation, and the alterations in electric potential landscapes in the active devices. Furthermore, we carried out selective peel-off-and-rebuild experiments and depth-profiling analyses to pinpoint the critical degradation site and reveal the underlying microscopic mechanism. The results indicate that the operation-induced efficiency increase results from the degradation of electron-injection capability at the electron-transport layer/cathode interface, which in turn leads to gradually improved charge balance. Our work provides new insights into the degradation of red quantum-dot light-emitting diodes and has far-reaching implications for the design of charge-injection interfaces in solution-processed light-emitting diodes.

Colloidal semiconductor quantum dots (QDs), such as II−VI QDs, III−V QDs, and metal-halide perovskite QDs, feature tunable bandgaps, narrowband emission, high photoluminescent quantum yields (PLQYs), and excellent solution processibility[1–5]. Quantum-dot light-emitting diodes (QLEDs) have emerged as cost-effective and high-performance electroluminescent (EL) devices for next-generation display and solid-state lighting technologies[6–8]. Recent years have witnessed substantial advances in the material chemistry of QDs[9–15], metal-oxide electron-transport layers (ETLs)[16–19], and organic hole-transport layers (HTLs) for QLEDs[20–22]. These efforts have led to red, green, and blue QLEDs with high external quantum efficiencies (EQE: >20%), low turn-on voltages, and high brightness[14,15,22–25]. At present, a major obstacle to the practical applications of QLEDs lies in their inferior operational lifetimes. Further understanding of the temporal

[1]Key Laboratory of Excited-State Materials of Zhejiang Province, State Key Laboratory of Silicon Materials, Department of Chemistry, Zhejiang University, Hangzhou, China. [2]State Key Laboratory of Modern Optical Instrumentation, College of Optical Science and Engineering, International Research Center for Advanced Photonics, Zhejiang University, Hangzhou, China. [3]Cavendish Laboratory, University of Cambridge, Cambridge, UK. [4]i-Lab, CAS Key Laboratory of Nanophotonic Materials and Devices, Suzhou Institute of Nano-Tech and Nano-Bionics, Chinese Academy of Sciences, Suzhou, China. [5]Center for Electron Microscopy, State Key Laboratory Breeding Base of Green Chemistry Synthesis Technology and College of Chemical Engineering, Zhejiang University of Technology, Hangzhou, China. ✉e-mail: yd359@cam.ac.uk; qchen2011@sinano.ac.cn; yizhengjin@zju.edu.cn

behavior and degradation mechanisms of QLEDs is urgently demanded from both technical and fundamental perspectives.

The operational stability of an LED is manifested by the changes in EL efficiency (or luminance at a constant current) under continuous operation. Typically, electrical stress on QLEDs causes continuous decreases in EL efficiencies[26,27], similar to the typical degradation behaviors observed in organic LEDs or inorganic LEDs[28,29]. The operation-induced efficiency losses of QLEDs have been attributed to the formation of defects in the HTLs[30-34], deterioration of the QDs[15,35,36], or increased exciton quenching by the ETLs[37].

In this work, we investigate an anomalous device-degradation phenomenon observed in red QLEDs, which features pronounced increases, instead of decreases, in the EL efficiencies upon continuous electrical excitations. As illustrated in Fig. 1, the efficiency of the QLED can increase by ~90% within the first 1,000 h of continuous operations. We note that similar phenomena have been observed in many previous reports of QLEDs[9,11,16,17,35,38-40], and some high-efficiency QLEDs need this pre-stressing process to reach an optimal operation condition. Unfortunately, the underlying mechanism remains to be explored. At the fundamental level, the significant enhancement in EL efficiency indicates that the status of the QLED at the beginning of the application lifespan (indicated by $L_{max}$) has largely deviated from that of the pristine device. In practice, this phenomenon causes unstable luminance that is technically undesirable. Furthermore, the nonmonotonic temporal evolution of luminance may cause ambiguity in the determination and comparison of the operational lifetime, adding a layer of complexity to the optimization of QLEDs. Therefore, it is crucial to understand the critical changes involved in the operation-induced efficiency elevation of the QLEDs.

We use shelf-stable red QLEDs as a model system to study the operation-induced efficiency elevation so that the storage-induced interferences can be excluded. It has been widely reported that QLEDs can show increased efficiencies and improved conductivities during the early stage of storage (first few days or few weeks)[18,41-45]. The so-called positive ageing effects are due to the in-situ reactions induced by the acidic resins used for device encapsulation. By using acid-free resin and optimizing the oxide ETLs, the positive ageing effects can be eliminated and shelf-stable QLEDs are obtained[15,18,46]. These shelf-stable QLEDs offer an ideal platform for investigating the operation-induced changes of the devices. We combine a series of *operando* characterization techniques to track the temporal evolutions of the charge dynamics during efficiency elevation and to image the spatial distributions of the electric potential landscapes before and after ageing. Furthermore, we combine the "selective peel-off and rebuilt" experiments and depth-profiling techniques on full stacks of QLEDs to identify the critical degradation site responsible for the efficiency-elevation effects.

## Results
### Efficiency elevation of the shelf-stable QLEDs
The operation-induced efficiency elevation is observed in the shelf-stable red QLEDs (Fig. 1a for the device structure). The QLED employs a widely-used architecture composed of poly((9,9-dioctylfluorenyl-2,7-diyl)-co-(4,4'-(N-(4-sec-butylphenyl)diphenylamine))) (TFB) as the HTL, CdSe-based core-shell QDs as the emissive layer, and highly-conductive ZnO nanoparticles (c-ZnO) as the ETL. The use of c-ZnO nanoparticles with a relatively large diameter of ~6 nm to deposit ETLs allows the formation of efficient electrical contact with the Ag cathode and offers decent electron transport in the ETLs. Acid-free epoxy resin is used to encapsulate the devices.

Under continuous operation, the QLED shows efficiency elevation that is long in time and large in amplitude (red curve, Fig. 1b). The luminance of the device increased by over 90% in ~1000 h of constant-current operation (10 mA cm⁻²). We note that the shelf-stable QLED shows negligible changes in EQEs during the storage (blue circles, Fig. 1b), verifying that the efficiency elevation does not originate from the positive ageing effects. The efficiency-elevation durations show a strong dependence on the driving current densities, e.g., from ~60 h at 30 mA cm⁻² to ~10 h at 100 mA cm⁻² (Fig. 1c and Supplementary Fig. 1).

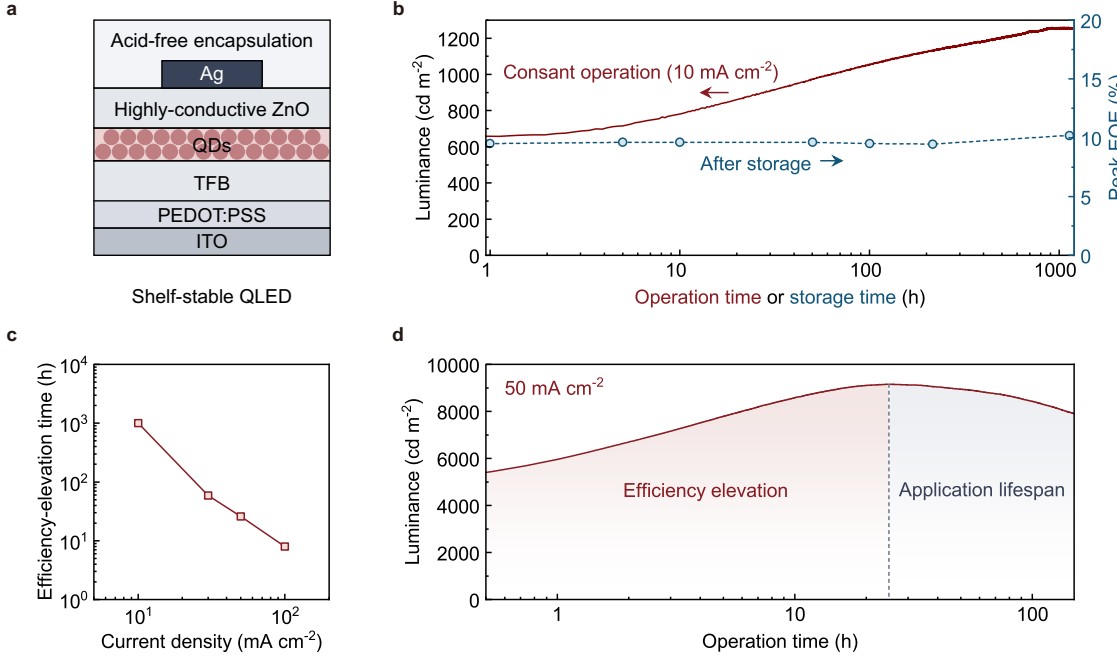

**Fig. 1 | Efficiency elevation of the shelf-stable QLEDs. a**, The device structure of the shelf-stable QLED. **b**, Time evolution of the luminance of the QLED driven by a constant current density of 10 mA cm⁻² (red curve). The control device shows minimal changes in the peak EQEs (blue circles) during storage. **c**, Dependence of the efficiency-elevation durations on the driving current densities. **d**, Time evolution of the luminance of the QLED in the accelerated lifetime test at 50 mA cm⁻². The device undergoes a long period of efficiency elevation (red-shaded region), prior to the luminance decay that determines the application lifespan (gray-shaded region).

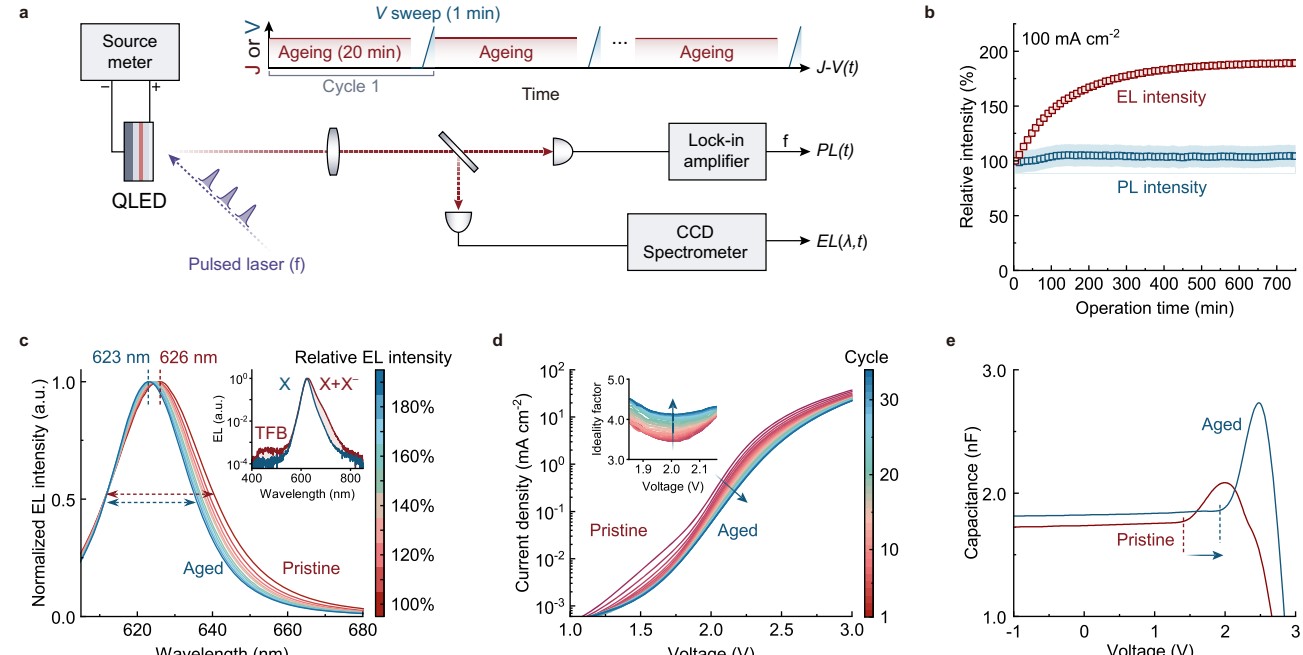

**Fig. 2 | Charge dynamics during efficiency elevation. a** Schematics of the experimental set-up for in-situ/operando characterizations on the efficiency elevation. The device is aged by a constant current density of 100 mA cm$^{-2}$. At regular intervals (red periods, top), a fast voltage sweep (blue periods, top) is employed to measure the $J-V$ characteristics of the device. During the electrical stressing, the EL spectra (EL($\lambda$,t)) are recorded by a CCD spectrometer. Meanwhile, the PL intensities (PL(t)) are obtained by exciting the QLED with a pulsed laser (103 kHz, 445 nm) and measuring the modulated responses with a lock-in amplifier. **b** Time evolutions of EL intensities (red squares) and PL intensities (blue circles) of the operating QLED.

The error bars for the measurements of the weak PL intensities are shown by the blue-shaded region. **c** Evolutions of the EL spectra during efficiency elevation from 100% (red) to 190% (blue) of the initial value. Inset: normalized EL spectra of the device before (red) and after the efficiency elevation (blue) plotted on a semi-log scale. **d** $J-V$ curves of the QLED measured at the intervals during the efficiency elevation. Inset: diode ideality factors ($\eta$) extracted from the $J-V$ curves. **e** $C-V$ characteristics of the device before (red curve) and after (blue curve) the efficiency elevation. The onset voltages for the capacitance rise are denoted by the dashed lines.

These results confirm that the increase in luminance is caused by electrical excitation.

According to an accelerated lifetime test (at 50 mA cm$^{-2}$, Fig. 1d), the efficiency elevation takes a considerable time (~20 h, red-shaded regime) prior to the general-defined application lifespan of the device (T$_{95}$: ~50 h, gray-shaded regime).

## Charge dynamics during efficiency elevation

We performed in-situ characterizations on an operating QLED during the efficiency-elevation process to comprehensively analyse the evolutions of both optical and electrical properties of the device. As illustrated by Fig. 2a (see Methods for details), the PL intensity of the QDs, the EL intensity, and the EL spectrum of a QLED were simultaneously monitored during the accelerated ageing at a constant current density of 100 mA cm$^{-2}$ (red periods). At regular intervals, fast voltage sweeps were imposed to extract the current-density-voltage (J-V) characteristics of the QLED (blue periods).

Optical measurements demonstrate that the efficiency elevation results from a higher conversion ratio of the injected carriers to the emissive neutral excitons (**X**) in the QDs ($\eta_X$, generation of excitons per injected electron). As shown by the evolutions of EL and PL intensities of the operating QLED (Fig. 2b), the increase of EL efficiency (by ~90%) is much larger than that of PL efficiency (by ~5%). This discrepancy suggests that the efficiency elevation in EL is not due to the suppression of exciton quenching, which would lead to identical changes in both EL and PL. Instead, the results indicate improvements in the exciton-generation efficiency, i.e., $\eta_X$. The evolution of EL spectra (Fig. 2c) displays a spectral narrowing (full width at half maxima, FWHM: from 30 nm to 24 nm) and blue-shift (peak wavelength: from 626 nm to 623 nm) after the efficiency elevation ("aged" device in Fig. 2). The final EL spectrum approaches that of **X** emission of the QD

film (Supplementary Fig. 2). It is known that the negatively-charged exciton (**X**$^-$) of CdSe-based QDs, which suffers from the rapid Auger recombination[47,48], features red-shifted emission with respect to that of **X**[49,50]. Accordingly, we suggest that both the bright state of **X** and the dim state of **X**$^-$ are electrically generated in the pristine QLED. The fraction of the electrical-generated **X**$^-$ is gradually decreased during the efficiency-elevation stage. Moreover, the parasitic emission from the TFB is reduced after the efficiency elevation (inset of Fig. 2c). This result indicates less electron leakage from QDs into TFB and thus less conversion of injected carriers to the excited states in HTL.

Electrical analyses show that the enhancement of $\eta_X$ is accompanied by the decreased charge-injection capability of the device. During the efficiency elevation, the slopes of the $J-V$ curves in the exponential regime continuously decline (Fig. 2d), indicating that larger voltages are required to provide efficient charge injection. Quantitatively, the empirical ideality factor ($\eta$) of a diode can be extracted from the exponential J-V characteristics according to the Shockley diode equation[51,52]:

$$\eta = \frac{e}{k_B T} \frac{\partial V}{\partial \ln J} \qquad (1)$$

where $e$ is the elementary charge, $k_B$ is the Boltzmann constant, and $T$ is the temperature. Literature reports on organic, inorganic, and perovskite LEDs suggest that the minimum value of $\eta$ in a $J-V$ curve, which corresponds to the steepest exponential incline in the exponential regime, correlates inversely with the charge-injection capacity of the device[53–56]. Our control experiments on the QLEDs with different cathodes (Supplementary Fig. 3) show that impeded electron injection at the cathode/ETL interface leads to an increase in $\eta$, consistent with the literature reports. As shown by the inset of

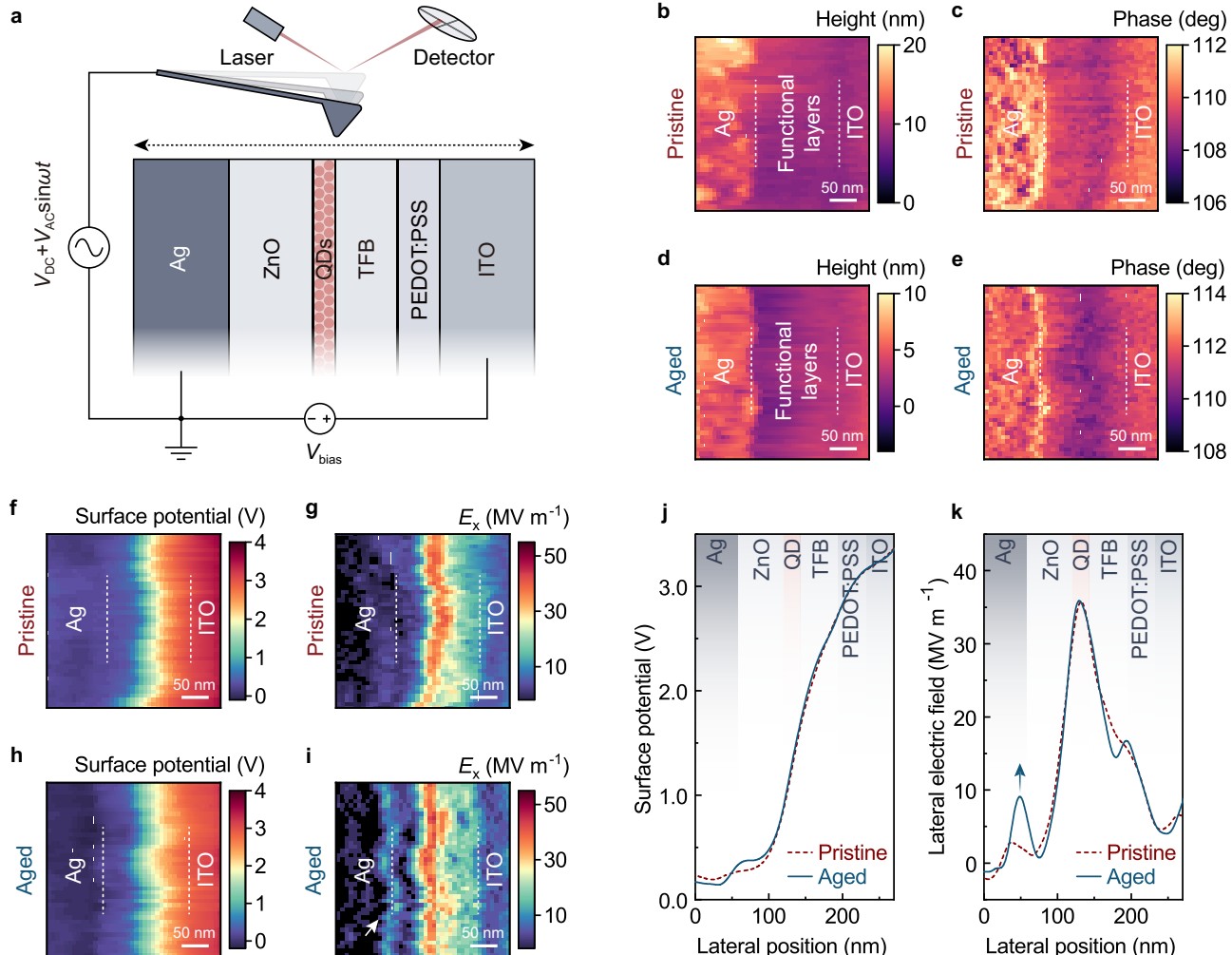

**Fig. 3 | Changes in the electric potential landscapes. a** Schematic diagram of the operando cross-sectional SKPM analysis. The surface potentials of the cross-sectional samples are scanned in charge-injection directions in the SKPM mode, while a constant bias ($V_{bias}$) is applied between the cathode and anode of the device. The height images **b**, **d**, phase images **c**, **e** of the pristine and aged devices, respectively. The surface potential images obtained from SKPM measurements **f**, **h** are used to extract the corresponding electric-field strengths in the lateral directions **g**, **i**. The interfaces between the electrodes and the functional layers are illustrated by dashed white lines. The overall thickness of the functional layers is consistent with the cross-sectional STEM measurement. **j**, Representative surface potential profiles of the pristine (red dashed curve) and aged (blue curve) under the same bias voltage of 3 V. **k** Electric-field distribution profiles of the pristine (red dashed curve) and aged (blue curve) devices extracted from **j**.

Fig. 2d, $\eta$ of the QLED shows an increase from 3.45 to 4.14 during the efficiency elevation. Furthermore, capacitance-voltage (CV) characteristics of the device before and after the efficiency-elevation process were examined (Fig. 2e). According to the previous studies, the capacitance rise in the CV curve of QLEDs using ZnO-based ETLs and organic HTLs represents the accumulation of electrons in the QD layer because of the more efficient transport and injection of electrons than those of holes in the devices[37]. The onset voltages for the capacitance rise of the pristine device and the aged device are 1.4 V (red dashed line) and 1.9 V (blue dashed line), respectively. Thus, the increased onset voltage of the CV curve implies that a larger driving force is required for electron accumulation in the QLED, indicating a decreased electron injection capability.

We note that the efficiency-elevation process is different from the temporal behaviors of the QLED at the initial stage upon turn-on. At the initial stage of the device operation (<10 min), the device shows rapid decreases in both the driving voltage and the luminance (Supplementary Fig. 4a), in consistency with some previous reports on transient behaviors of QLEDs[30,32,57]. Subsequently, both the driving voltage and luminance gradually increase in the long-term efficiency-elevation

process. In contrast to the temporal evolutions that repeatedly occur regardless of the aging stage, the efficiency elevation is due to irreversible change in the steady-state properties of the QLED (Supplementary Fig. 4b).

## Changes in the electric potential landscapes

To spatially resolve the changes in electrical properties inside the multilayer devices, we applied cross-sectional scanning Kelvin-Probe microscope (SKPM) measurements on operational QLEDs (Fig. 3a). The operando cross-sectional SKPM is a powerful scanning-probe technique that has been successfully applied in several thin-film solar cells and photodetectors to image their electric potential landscapes during operation[58-61]. In our experiments, cross-sectional samples of a pristine QLED and an aged QLED (after the efficiency-elevation process) were prepared via ion-beam milling (see Methods for details). J–V characterizations on the cross-sectional devices demonstrate that the electrical properties of the pristine and the aged devices are preserved (Supplementary Fig. 5).

Figure 3b–e shows the height images and the corresponding phase images of the pristine and aged QLED, respectively.

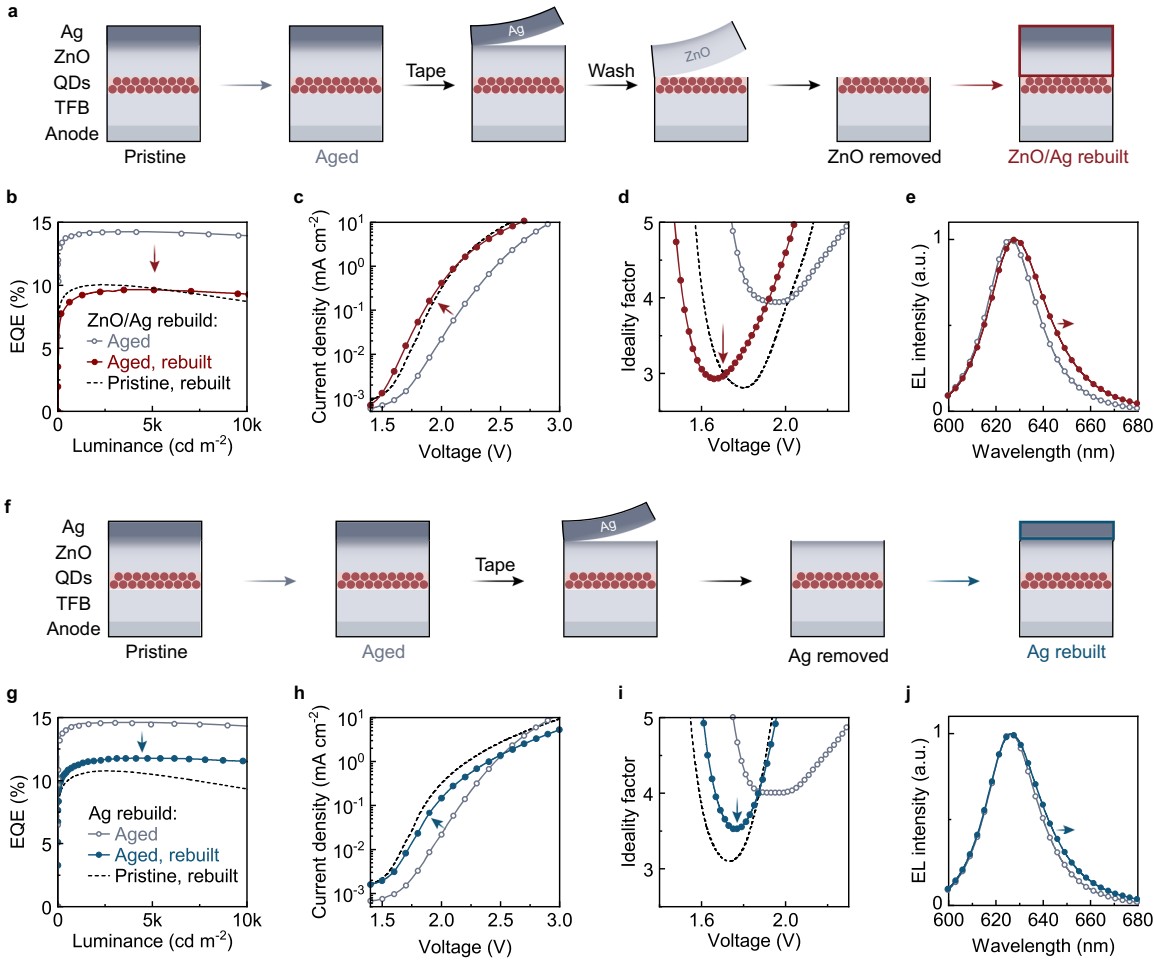

**Fig. 4 | Pinpoint the degradation site. a** Schematics of the peel-off-rebuild experiments to rebuild the Ag cathode and the ZnO ETL from the aged QLED. The EQEs **b**, J–V characteristics **c**, ideality factors **d**, and EL spectra **e** of the aged device (gray hallow circles) are shown in comparison with the aged device after rebuilding the ZnO/Ag structure (Methods). The properties of the pristine device with a rebuilt ZnO/Ag structure are represented by the gray dashed curves. **f**, Schematics of the peel-off-rebuild experiments to rebuild the Ag cathode from the aged QLED. The EQEs **g**, J–V characteristics **h**, ideality factors **i**, and EL spectra **j** of the aged device (gray hallow circles) are shown in comparison with the aged device after rebuilding the Ag cathode. The properties of the pristine device with a rebuilt Ag cathode are represented by the gray dashed curves.

The high-quality QLED cross-sections with small fluctuations in the height channels and clear contrasts in the phase channels allow us to identify the different regions of electrodes and functional layers (interfaces indicated by the dashed lines). The surface potential images of the pristine and aged QLEDs under the same bias voltage of 3 V are shown in Fig. 3f and Fig. 3h, respectively. The partial derivatives of the surface potentials with respect to their lateral positions, which represent the electric fields normal to the charge-injection directions in the QLEDs, are extracted (Fig. 3g, i). Notably, a more rapid rise in the surface potential at the interface near Ag is observed in the aged device, which corresponds to an emerged strong electric field region near the ZnO/Ag interface in the electric-field images (indicated by the arrow in Fig. 3i). These features can be more evident in the comparisons of the representative profiles of the surface potential distributions (Fig. 3j) and the lateral electric-field distributions (Fig. 3k) in the pristine and aged QLEDs.

The above experiments imply that the decreased charge-injection capability of the QLED after efficiency elevation is primarily due to the additional voltage drop and thus an increased resistance at the ZnO/Ag interface. We also note that there are only minor changes in the electric potential and electric field distributions in the other functional layers.

## Pinpoint the degradation site

Based on the above clues obtained from operando measurements, we designed the "selective peel-off and rebuild" experiments on aged QLEDs to pinpoint the site responsible for the efficiency-elevation phenomenon. Specifically, we examined whether an aged QLED can be restored to its pristine status by selectively regenerating some of the functional layers. The cathode and the ETL of the aged QLED can be rebuilt by removing the Ag layer, washing off the ZnO nanoparticles, and then depositing fresh materials (Fig. 4a). To rule out additional interferences from such processing, pristine devices were also subjected to the peel-off and rebuild processes in the control experiments.

Comparisons of the optoelectronic characteristics of the rebuilt QLEDs and the pristine devices suggest that the degradation of the ZnO/Ag interface accounts for the operation-induced efficiency elevation. By rebuilding the ZnO and Ag layers from an aged device, the device characteristics, including EQE (Fig. 4b), J–V characteristics (Fig. 4c), ideality factors (Fig. 4d), and EL spectrum (Fig. 4e), recover from those of the aged status after efficiency elevation (gray hallow circles), approaching those of the pristine device (dashed curves). Furthermore, by selectively rebuilding the Ag cathode (Fig. 4f), the device characteristics (Fig. 4g–j, blue solid circles) are largely recovered from the aged status. We note that the incomplete recovery of the

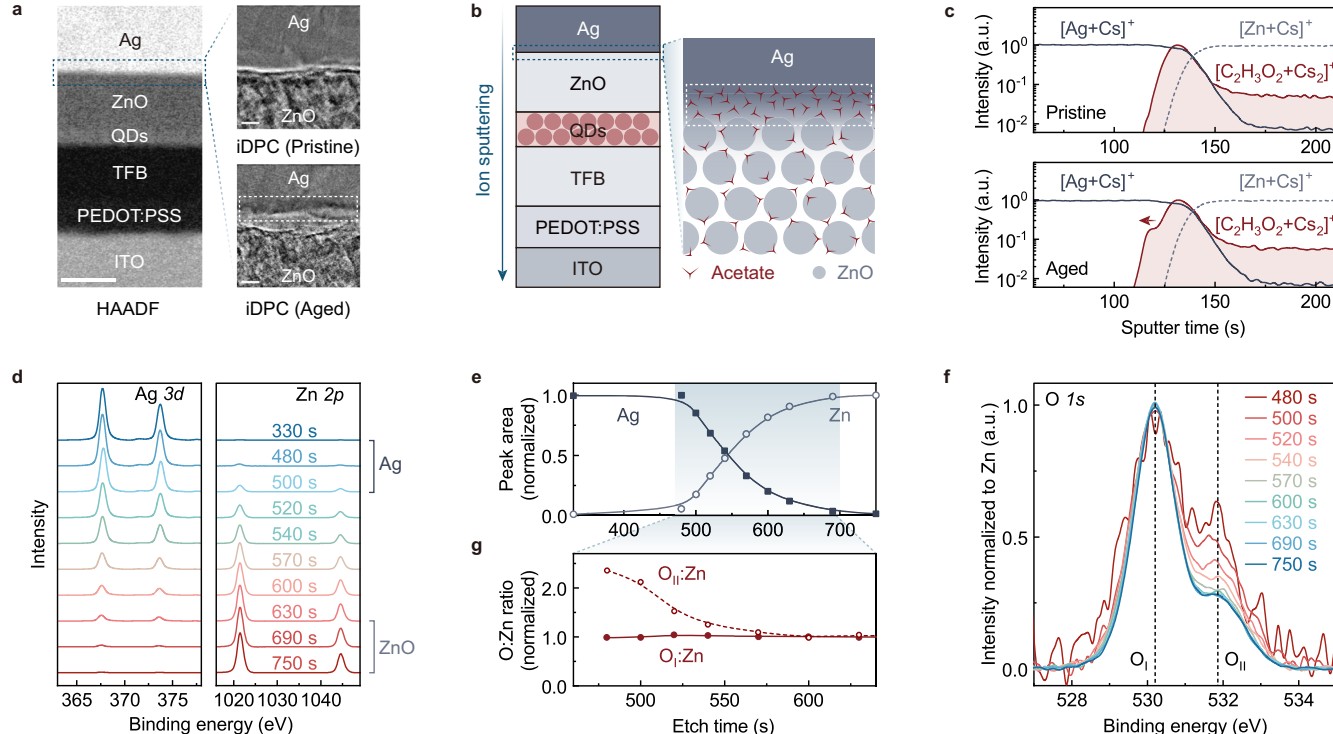

**Fig. 5 | Depth profiling of chemical compositions. a** HAADF-STEM image (left, scale bar: 50 nm) of the cross-section of a QLED. The iDPC-STEM images zooming in at the Ag/Zn interfaces of the pristine (top right) and aged (bottom right) devices are shown for comparison (scale bar: 5 nm). **b** Schematic diagram of the depth profiling analysis of the QLED (left), and the resolved chemical compositions at the ZnO/Ag interface (right). **c** Normalized ToF-SIMS profiles of [Ag+Cs]$^+$ (gray solid curves), [Zn+Cs]$^+$ (gray dashed curves) and [C$_2$H$_3$O$_2$ + Cs$_2$]$^+$ (red solid curves) at the ZnO/Ag interface of the pristine device (top) or the aged device (bottom). The acetates spread into the Ag (red arrow, bottom) after efficiency elevation. **d** XPS core-level spectra for Ag $3d$ (left) and Zn $2p$ (right) at different depths in the device. The corresponding etching times before acquiring the XPS spectra are labeled. **e** Relative atomic numbers of Ag and Zn extracted from the peak areas of the XPS spectra. **f** Depth evolutions of the O $1s$ spectra at the ZnO/Ag interface, which are normalized according to the atomic numbers of Zn (**e** top) at the corresponding depths. The binding energies for the lattice oxygen (O$_I$) and the surface oxygen species (O$_{II}$) are denoted by dashed lines. **g** Normalized atomic ratios of O$_I$:Zn (red solid circles) and O$_{II}$:Zn (red hollow circles) at the ZnO/Ag interface.

Ag-rebuilt device may be due to the residue of aged species at the ZnO/Ag interface that cannot be fully removed by the tape (Supplementary Fig. 6).

## Structural and chemical analyses

Scanning transmission electron microscopy (STEM) techniques were utilized to directly observe the structural alterations at the ZnO/Ag interface after efficiency elevation. Figure 5a shows the high-angle annular dark-field (HAADF) images of the device cross-section. The ZnO/Ag interfaces of the pristine (top right) and the aged QLEDs (bottom right) were imaged by using the integrated differential phase contrast (iDPC) mode. The iDPC-STEM enables simultaneous imaging of both light and heavy atoms[62,63], which is suitable for revealing the contrast at the organic-inorganic hybrid interface. Compared with the relatively smooth and flat interfacial region in the pristine device, a "trench" (manifests as an undulation of contrast) appears on the Ag side of the ZnO/Ag interface (white dashed square).

Depth-profiling characterizations were further carried-out to probe the chemical compositions at the ZnO/Ag interface. In the time-of-flight secondary ion mass spectrometry (ToF-SIMS) measurements, a focused ion beam (**Cs**$^+$) continuously sputters the surface species (**M**) of the QLED. The chemical compositions are profiled from the electrode to the functional layers inside the device (Fig. 5b, left) by detecting the signals from the corresponding cluster ions ([**M**+**Cs$_n$**]$^+$) after different sputter times. The normalized ToF-SIMS profiles at the ZnO/Ag interface, including the depth distributions of the Ag segments (solid gray curve), the Zn segments (dashed gray curve), and the acetate segments (red curve), are shown in Fig. 5c. Remarkably, in

addition to the distributions of acetate within the ZnO layer as surface ligands of nanocrystals, there is an accumulation of acetates at the ZnO/Ag interface in the pristine device (top of Fig. 5c). After efficiency elevation, the accumulated acetates at the interface further spread into the Ag layer (red arrow, bottom of Fig. 5c), while signals of other species remain unaffected in the aged device (Supplementary Fig. 7). It is also noteworthy that the average concentration of acetate in the ZnO film is about ~1.8 × 10$^{-3}$ mol cm$^{-3}$, which is well above the detection limit of ToF-SIMS (Supplementary Fig. 8). Further measurements on the depth profiles of the ZnO/Ag interfaces in QLEDs subjected to various ageing time confirm the trend of acetate migration into the Ag electrodes (Supplementary Fig. 9).

The interfacial accumulation of acetates is confirmed by X-ray photoelectron spectroscopy (XPS) depth profiling. Figure 5d shows the Ag $3d$ and Zn $2p$ core-level spectra of the device after different etching times by the Ar$^+$ ion. The ZnO/Ag interface is probed during the etching time of 480–690 s, where the Ag signals decrease and the Zn signals increase. The spectral evolutions of the O $1s$ spectra are shown in Fig. 5f, which are composed of a low-binding-energy O component (530.2 eV, O$_I$) from the O$^{2-}$ ions in the ZnO lattice, a middle-binding-energy constituent from O$^{2-}$ ions near oxygen vacancies (531.6 eV, O$_{III}$) and a high-binding-energy peak (532.0 eV, O$_{II}$)[41,64–66] from the oxygen species on the surface of ZnO nanoparticles (see Supplementary Fig. 10 for Gaussian fits). The intensity changes of the lattice O$_I$ are in parallel with those of the Zn$^{2+}$, as demonstrated by the constant atomic ratio of O$_I$:Zn in the ZnO layer (solid red circles, Fig. 5g). In contrast, there is a higher relative content of the surface O$_I$ (O$_{II}$:Zn ratio) close to the Ag contact comparing with that inside the ZnO layer (hallow red

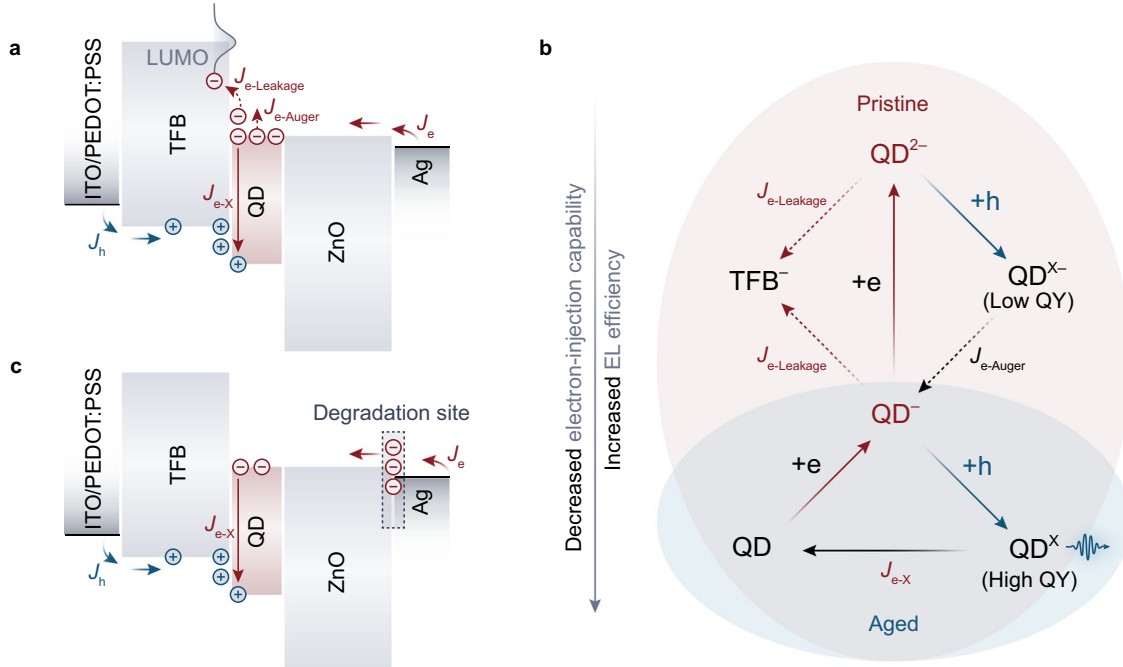

**Fig. 6 | Mechanism of the operation-induced efficiency elevation. a** Flat-band energy-level diagrams of the pristine QLEDs. The current flows (arrows) and space-charge accumulations (symbols) of electrons (red) and holes (blue) in the QLED under steady-state operation are illustrated. The high space-charge concentration of electrons in the QD layer results in considerable consumptions of $J_e$ via the efficiency-loss channels of non-radiative Auger recombination ($J_{e\text{-Auger}}$) and electron leakage ($J_{e\text{-leakage}}$). Consequently, the pristine devices possess low exciton-generation efficiency ($\eta_X$), i.e., the conversion ratio of electron current into exciton-recombination current ($J_{e\text{-X}}$), **b** Schematic diagram summarizing the EL process in the QDs before (red shaded) and after the efficiency elevation (blue shaded). As the electron-injection capability degrades, the EL process is gradually dominated by the electrical generation and recombination of the highly emissive single exciton state ($QD^X$, top right). **c** Schematics illustrating the distributions of charge carriers and current flows in the QLED after the efficiency-elevation process. More electrons are accumulated at the degradation site (dashed square). The contributions of $J_{e\text{-Auger}}$ and $J_{e\text{-leakage}}$ channels are suppressed due to the decreased electron concentration in the QD layer (compared with **a**), which leads to a higher conversion ratio of the electrons into exciton recombination.

circles, Fig. 5g). The results are in line with the accumulation of excessive acetates at the ZnO/Ag interface.

For a more rigorous assignment of the degradation site, we conducted a series of comparisons of the structural and optical properties of other functional layers before and after the efficiency-elevation process. The TFB HTL shows minimal changes in the PL spectra after efficiency-elevation process (Supplementary Fig. 11). Meanwhile, the PEDOT:PSS HIL shows nearly identical Raman spectra (Supplementary Fig. 12). XPS analyses on the bulk part of the ZnO ETL in the aged QLED show no discernible changes in the chemical status of Zn and O elements (Supplementary Fig. 13), excluding the role of oxygen-related defects in the efficiency-elevation process. This is consistent with the absence of resistance hysteresis of our QLEDs (Supplementary Fig. 14), which is a common signal of oxygen-vacancy migration in literature reports[66–68]. Based on the above results, we conclude that the critical degradation site is the acetate-accumulated region adjacent to the ZnO/Ag interface.

## Discussion

We summarize the key changes of charge dynamics that account for the efficiency elevation of the QLED (Fig. 6). In the pristine QLED, the high electron mobility of the c-ZnO layer and the efficient electrical contact of the c-ZnO/Ag lead to significantly higher space-charge concentration of electrons in the QD layer comparing with that of holes (Fig. 6a). From a microscopic perspective (Fig. 6b), the electron injection to negatively-charged QD ($QD^- \xrightarrow{+e} QD^{2-}$) in this device can compete with the hole injection into the QD at the HTL/QD interface ($QD^- \xrightarrow{+h} QD^X$) in the condition of high electron concentration in the QD layer. The $QD^{2-}$ states are converted to the dim state of $X^-$ upon hole injection ($QD^{2-} \xrightarrow{+h} QD^{X-}$, Fig. 6b), which eventually consumes a

faction of the injected electron currents ($J_e$) into the non-radiative recombination currents via Auger recombination ($J_{e\text{-Auger}}$, dashed arrow in Fig. 6a). Meanwhile, the high concentration of electrons in the QD layer enhances the probability of electron transfer to the tail states in the LUMO of the TFB[22,50,69,70], causing another channel of efficiency loss in terms of electron leakage out of the QD layer ($J_{e\text{-leakage}}$, dashed arrow in Fig. 6a). Consequently, the exciton-generation efficiency ($\eta_X$), defined by the conversion ratio of injected electrons to excitons ($\eta_X = J_{e\text{-X}}/J_e$), is relatively low in the pristine device.

As the electron-injection capability decreases, the space-charge distributions across the QLED gradually evolve in compliance with the Poisson equation. In the aged QLED, due to the increased resistance at the ZnO/Ag interface, more space charges of electrons are accumulated at the electron-injection interface and less are accumulated in the QD layer (Fig. 6c) compared with the status in the pristine device (Fig. 6a). The reduced electron concentration in the QD layer leads to decreased conversion ratio from injected electrons to $X^-$ ($J_{e\text{-Auger}}/J_e$) or leakage current in TFB ($J_{e\text{-Leakage}}/J_e$). Consequently, at a constant total electron current, a higher ratio ($\eta_X$) of the injected electrons is converted into the desired bright state of $X$ (bottom, Fig. 6b). Therefore, the operational degradation of the device, which was generally presumed to be detrimental to LEDs, can in turn promote the electrical-excitation efficiency in our highly-conductive and shelf-stable QLEDs.

We also highlight the critical impacts of the insulating organic impurities on the stability of the electrical contact between the ZnO ETL and metal cathode. We suspect that the accumulated acetates may have originated from the weakly bound ligands or residual precursors in the dispersion of ZnO nanocrystals. During the spin-coating process, the excessive acetate-containing species precipitated from the solution, forming an interlayer at the top of the ZnO-based ETLs. After the

deposition of the metal cathodes, an unstable contact at the interface occurs. The migration and diffusion of the excessive acetates during the device operation (bottom of Fig. 5c) would cause degradation of electrical contact, leading to the gradual decrease of the electron-injection capability. Therefore, the efficiency-elevation phenomenon, which may seem like an improvement in the EL performance at first glance, is actually due to the continuous deterioration of the cathode contacts by the unwanted impurities. In long-term operations, these processes might eventually contribute to more severe device degradation.

While the above findings suggest an inherent instability of the ZnO/Ag interface, we highlight that whether its degradation would lead to the efficiency elevation of a QLED depends on two factors. First, it depends on the initial status of a QLED. An essential reason that the deterioration of the ZnO/Ag interface can lead to efficiency elevation in our shelf-stable red QLED is that the pristine operational device is in a status with excessive distribution of electrons in the QD layer. The low initial $\eta_X$ can be increased upon the degradation of the electron-injection capability. As for a hypothetical device with near-unity internal quantum efficiency and $\eta_X$, the degradation of the electron-injection capability would unlikely further improve the efficiency. Second, the improvement in the $\eta_X$ in our devices indicates an unparalleled degradation of the bipolar injection/transport properties, i.e., a slower degradation rate of hole-injection capability than that of the electron-injection capability during the efficiency-elevation stage. For other QLEDs with unstable HTLs, the degradation behaviors could be governed by multiple channels and might not readily lead to an increased $\eta_X$. Our QLED with the highly conductive ZnO ETL and the relatively stable HTL provides a unique platform to observe the degradation effects of the ZnO/Ag interface that could be hidden in other studies.

In short, we have reported an anomalous operational degradation pattern of red QLEDs that electrical excitations induce a continuous elevation of EL efficiency. This results from the continuous degradation of electron-injection capability in device operation, which in turn reduces the conversion of injected carriers into the low-efficiency states of $X^-$ and leakage currents out of the QD layer. We have pinpointed the critical degradation site to be the ZnO/Ag interface, where there is an accumulation of excessive acetates that deteriorate the cathode contact during device operation. Our work provides new insights into the degradation mechanism of state-of-the-art red QLEDs, highlighting the critical impacts of the impurities at the electrode contact on the device stability. Given that hydrolysis of zinc acetates is the predominant synthetic method for ZnO nanoparticles used in the state-of-the-art QLEDs[11–15,22,71–73], acetate migration at the ETL/cathode interface could be general in these devices. Future efforts on the purification of ZnO-nanoparticle dispersions or surface modification of the solution-processed ZnO ETLs should be helpful to stabilize the electrical contacts.

We also note that ZnO nanoparticles as well as ZnO/cathode interfaces are widely employed in solution-processed optoelectronic devices, including Cd-free QLEDs[12], perovskite LEDs[74,75], photodetectors[76,77], quantum-dot solar cells[78,79], and organic or perovskite solar cells[80–83]. We expect that our work shall provide broad implications to understand the degradation mechanisms of these solution-processed optoelectronic devices. Moreover, our diverse and complementary methods for QLED characterizations could be extended to investigate the degradation mechanisms in other devices.

## Methods

### Materials
TFB (average molecular weight, ~90,000 g mol⁻¹) was purchased from American Dye Source. Colloidal red QDs (CdSe/CdZnSe/ZnSeS) were provided by Nanjing Technology Co., Ltd. Lithium hydroxide (≥ 98%) and zinc acetate dihydrate (≥ 99.0%) were purchased from Sigma-Aldrich. Chlorobenzene (super dry, 99.8%), octane (super dry, 99%),

ethanol (super dry, 99.5%), ethyl acetate (HPLC grade, 99.9%), and dimethyl sulfoxide (DMSO, HPLC grade, 99.9%) were purchased from J&K Chemical Ltd.

### Synthesis of the highly conductive ZnO nanoparticles
A solution of 1.5 mmol $Zn(CH_3COO)_2$ in 15 mL dimethyl sulfoxide (DMSO, HPLC grade, 99.9%; J&K Chemical Ltd.) was mixed with a solution of LiOH (2.5 mmol) in ethanol (25 mL). The mixture was stirred for 2 h at 50 °C under ambient conditions. The ZnO nanoparticles were precipitated by adding ethyl acetate and further purified by dispersing/precipitating twice using a combination of ethanol/ethyl acetate. The resultant oxide nanocrystals were redissolved in ethanol and filtered before use.

### Device fabrication
PEDOT:PSS solutions (in water, Clevious PVP Al 4083, filtered through a 0.22 μm N66 filter) were spin-coated onto ITO-coated glass substrates (sheet resistance: ~20 Ω sq⁻¹) at 3000 r.p.m. for 40 s and baked at 150 °C for 30 min in air. TFB (in chlorobenzene, 12 mg mL⁻¹), QDs (in octane, 15 mg mL⁻¹), and c-ZnO nanoparticles (in ethanol, 30 mg mL⁻¹) were layer-by-layer deposited by spin coating at 2000 r.p.m. for 40 s. The TFB layers were baked at 150 °C for 30 min before the deposition of the QD layers. Next, silver electrodes (120 nm) were deposited through a shadow mask using a thermal evaporation system (Trovato 300 C) under a high vacuum (<10⁻⁶ Torr). The device area defined by the overlapping of the ITO and silver electrodes is 4 mm². Acid-free UV-curable resins (LOCTITE 3335) were used to encapsulate the devices. Finally, the devices were annealed at 85 °C for 30 min before the characterizations.

### Characterizations of QLEDs
The efficiency characterizations were performed in a nitrogen-filled glovebox ($O_2 < 1$ p.p.m., $H_2O < 1$ p.p.m.). A home-build system consisting of a source meter (Keithley 2400, Tektronix) and an integration sphere (FOIS-1, Ocean Optics) coupled with a CCD spectrometer (QE-Pro, Ocean Optics) was used to obtain the current-luminance-voltage curves. The operational lifetimes of QLEDs were measured by using an ageing system with an embedded photodiode designed by Guangzhou New Vision Opto-Electronic Technology Co., Ltd. The capacitances were measured by using a Precision LCR Meter (TH2838, Changzhou Tonghui Electronic Co., Ltd). The amplitude and frequency of the oscillating signal were set to 50 mV and 1000 Hz, respectively. The cross-sectional samples used for STEM characterizations were prepared by using a dual-beam focused-ion-beam system (Quata 3D FEG). The samples were analyzed by a transmission electron microscope equipped with double (image and probe) spherical aberration (Cs) correctors (Thermo Scientific Spectra 300).

### In-situ characterizations of the operating QLED
The QLED was driven by periodical electrical stressing (top, Fig. 2a) controlled by a Labview program. In each cycle, the device was subjected to a constant current density (100 mA cm⁻²) for 20 min, followed by a 0 V bias for 1 min, and then a voltage sweep (0–3.8 V) for 50 s. In the meantime, the active area of the QLED was optically excited by a focused pulsed laser (445 nm) with a repetition frequency of 103 kHz. The excitation power was kept low to ensure that the PL intensity is <0.01% of the EL intensity, excluding the interferences of optical excitations on the device operation. The EL spectra were collected by a CCD spectrometer (Ocean Optics) and the PL intensities were monitored by a photodetector (PDA100A, Thorlabs) coupled to a lock-in amplifier (SR830, Stanford Research Systems Inc.).

### Operando cross-sectional SKPM
To prepare a smooth cross-sectional sample, a device was cut from the back of the glass substrate and transferred into the vacuum chamber

(6.4 × 10⁻⁵ Torr) of an Ilion⁺ 693 System (Gatan Inc.). The device was cooled using liquid nitrogen and the exposed edge was milled by Ar ions using a beam voltage of 5 keV and a beam current of 10 µA for about 2 h.

The surface potential images of cross-sectional devices were performed using a Cypher S AFM (Asylum Research, Oxford Instruments) combined with a HF2LI Lock-in amplifier (Zurich Instrument) installed in an Ar-filled glove box. A single-pass scan mode was employed for frequency-modulation (FM) SKPM using a conductive AFM tip (NSC14 Cr/Au, Mikromasch) with a resonance frequency ω of ~140 kHz and a spring constant of ~5.0 N m⁻¹. While performing a standard AC mode scan to acquire the topography and phase signal of the sample, an AC voltage (typically 3 V in amplitude and 1 kHz in frequency) and a DC voltage were applied to the tip. The DC voltage (range from −10 V to +10 V) tuned by a homemade Kelvin controller that nullifies the amplitude of electrostatic force gradient sensitive sidebands at ω ± 1 kHz was collected as the surface potential signal. During the operando SKPM measurements, the bias voltages of devices were applied via a homemade tunable voltage source. The representative profiles of the surface potentials were obtained by averaging 5-line scans in the SKPM results.

## Peel-off and rebuild experiments
The peel-off-rebuild experiments were carried out on unencapsulated QLEDs in a nitrogen-filled glovebox. The electrical ageing before the peel-off and rebuild processing was carried out on a lifetime test system installed in a glovebox. The Ag cathodes were removed by using polyimide tapes. The ZnO ETLs were washed off by spin-coating (2000 r.p.m for 40 s) 30 µL of ethanol solution of acetic acid (volume fraction: 0.5%) and 30 µL of ethanol onto the Ag-removed devices.

## Depth profiling characterizations
The ToF-SIMS characterizations were carried out on TOF.SIMS5-100 (IONTOF) in an ultrahigh vacuum chamber (<3 × 10⁻¹⁰ mbar). A Cs⁺ ion beam (500 eV) was used as the sputter species, and a Bi⁺ ion beam was used as the primary ion projectile. The mass resolution is m/Δm >15,000 and the depth resolution is <1 nm according to the slow and steady sputtering rate of ~1 µm h⁻¹ with the data-acquisition rate of >30 scans min⁻¹. The limit of detection of the ToF-SIMS is on the order of 10⁻⁹–10⁻⁶ (ppb to ppm) in terms of concentration in the thin film. The **MCs⁺** approach was adopted to analyze the [**M** + **Cs**]⁺ or [**M** + **Cs₂**]⁺ cluster ions (e.g. m/z ratios of 196.83, 239.81, and 324.84 corresponding to [Zn+Cs]⁺, [Ag+Cs]⁺, and [C₂H₃O₂ + Cs₂]⁺, respectively), the depth profiles of which were corrected by those of Cs⁺ or Cs₂⁺ ions respectively. In the XPS depth profiling, an Ar⁺ ion beam (1000 eV) was used to etch the sample. The XPS spectra were acquired on Thermo Scientific ESCALAB Xi⁺ spectrometer in an ultrahigh vacuum with an Al kα radiation source.

## Raman spectroscopy and Fourier transform infrared spectroscopy (FTIR)
The Ag cathodes and the ZnO ETLs of the samples used for Raman characterizations were peeled off sequentially under the same condition mentioned above. The QDs and TFB layers were washed off by spin-coating (2000 r.p.m for 40 s) 30 µL chlorobenzene. Raman spectra were collected by a home-built microscope setup, in which a 532 nm CW laser and an InGaAs detector (PyLon IR 1700, Princeton Instrument) were used. To quantitatively analyse the absolute concentration of acetate within the ZnO film, a ZnO film (60 nm) deposited on an Au mirror was measured by a Nicolet iS50 Spectrometer (Thermo Fisher Scientific) in the reflection mode.

## Data availability
The data that support the findings of this study are available within the article and its Supplementary Information. All other relevant data are available from the corresponding authors upon request.

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

## Acknowledgements
We thank the technical support of Nano-X from Suzhou Institute of Nano-Tech and Nano-Bionics, Chinese Academy of Sciences (SINANO). We thank Dr R. Huang (SINANO) and Ms Z. Li (SINANO) for the ToF-SIMS and XPS characterizations. We acknowledge financial support from the National Key R&D Program of China (2021YFB3602703 and 2022YFB3606503 (Y.J.); 2021YFA1202802 (Q.C.)), National Natural Science Foundation of China (21975220 (Y.J.); 22022205 and 22372193 (Q.C.)), Key Research and Development Program of Zhejiang Province (2020C01001 (Y.J.)), China Postdoctoral Science Foundation (2021M702800 (Y.D.)) and the CAS Project for Young Scientists in Basic Research (YSBR-054 (Q.C.)).

## Author contributions
Y.D. and Y.J. conceived the idea and supervised the work. S.H. and Y.D. designed the experiments. S.H. fabricated the QLEDs, conducted the lifetime characterizations, performed the in-situ characterizations, and developed the peel-off-rebuild experiments. Y.D. improved the optical measurements and assisted in the data analysis. X.T., W.J., X.L., and C.W. assisted in the fabrication of QLEDs. D.C. assisted in the synthesis of ZnO nanoparticles. T. S. conducted cross-sectional STEM characterizations. N. Y. and Q. C. performed operando cross-sectional SKPM measurements, analyzed the data, and participated in revisions. Y.D. and Y.J. wrote the first draft of the manuscript and provided major revisions. All authors discussed the results and commented on the manuscript.

## Competing interests
The authors declare no competing interests.
