## [Peer Review File · Nature Communications]

REVIEWER COMMENTS

Reviewer #1 (Remarks to the Author):

In this work, the author claims that the anomalous efficiency elevation of QLEDs is induced by the degradation of the electron-transport layer/cathode interface. This conclusion is proved by using In-situ/operando characterizations, peel-off-and-rebuild experiments, and depth-profiling analyses. However, the author still needs to solve several key problems.

1. The efficiency elevation of QLEDs is a common phenomenon and has been observed in many reports. The author claims the anomalous efficiency elevation of QLEDs is induced by the interface of ZnO ETL and Ag electrode. However, the anomalous efficiency elevation of QLEDs was observed regardless of whether the ZnO electron transport layer was included in the device (ACS Nano, 2022, 16, 6, 9631-9639; Adv. Mater. 34, 4, 2106276). Therefore, it is unconvincing that the author think the anomalous efficiency elevation is completely induced by the interface of ZnO ETL and Ag electrode. The author must strictly prove whether the remaining functional layers and interfaces are unchanged, rather than just monitoring the PL of the QDs.

2. The EQE is only 10% when the author studies the anomalous efficiency elevation of the device. However, the author has achieved EQE of 20% of the shelf-stable QLEDs in 2020 (Adv. Mater. 2020, 32, 52, 2006178). When the EQE exceeding 20%, the device does not show the anomalous efficiency elevation (Adv. Mater. 2020, 32, 52, 2006178, Figure 5e)?

3. The in-situ/operando characterizations, peel-off-and-rebuild experiments have been reported many times (Nat. Photonics, 2022, 16, 505-511; Nat. Commun. 2020, 11, 2309; J. Phys. Chem. Lett. 2020, 11, 12, 4649-4654.). It will be necessary and important for the authors to clearly demonstrate the uniqueness and innovation compared to previous methods.

4. Compared with a fresh device, a red-shifted EL spectrum was observed in aged devices. The author believes that this is due to the fraction of the electrical-generated X⁻ is gradually decreased during the efficiency-elevation stage. This explanation is hard to understand. Even during the efficiency-elevation stage, the injection rate of electrons will decrease, but it should still be higher than the hole injection rate. Therefore, more and more electrons should be accumulated in QDs, which will lead to increased X⁻.

5. Previous work believed that the leakage of electrons into TFB during operation will worsen the hole injection speed (J. Phys. Chem. Lett. 2020, 11, 12, 4649-4654.). Is the deterioration of hole injection rate slower than that of electron injection rate during efficiency-elevation stage?

6. The author think that the incomplete recovery of the Ag-rebuilt device may be due to the residue of aged species at the ZnO/Ag interface. Is it possible that the electron injection speed was also reduced at

the interface of ZnO and QDs? Due to the electron injection speed was reduced at the interface of ZnO and QDs, resulting in the EQE of aged Ag rebuilt device is higher than that of the fresh Ag rebuilt.

7. The author believes that the reduced electron injection speed at the interface of ZnO and Ag is due to is due to the electrical contact degradation caused by the diffusion of Ac^- to the Ag electrode. Please provide some direct proof.

8. Please check the manuscript carefully. The $\text{C}_2\text{H}_3\text{O}_2^+$ in the Figure 4b should be changed to $\text{C}_2\text{H}_3\text{O}_2^-$.

Reviewer #2 (Remarks to the Author):

This manuscript presents a detailed analysis of the microscale physical processes responsible for the anomalous increase in luminance observed during the aging of quantum dot light-emitting diodes (QLEDs). The authors conclude that the primary cause of this phenomenon is the migration of ligands in the metal oxide nanocrystal layer, which leads to degradation at the ZnO/Ag interface. The presented argument is supported by a series of verification experiments, leading to a logical and self-consistent narrative.

It is important to note, however, that the mechanisms underlying device degradation are complex, and the model proposed in this study may not fully capture all relevant factors. Compared to the experimental skills and characterization, the discussion of this paper is weak.

To ensure the manuscript's high-quality and enhance its impact, the reviewer suggests that authors address the following questions and provide a more comprehensive discussion, then resubmitted.

Comment 1:

As discussed in the introduction section, similar phenomena have been widely reported in previous literature. Under current stress, the efficiency of quantum dot light-emitting diodes (QLED) is promoted. Some previous reports have discussed the behavior and attributed it to electrical unstable oxygen-related defects in ZnO-based nanocrystals. In this study, the electrical aging behavior of QLED is analyzed, and these factors cannot be overlooked. Specifically, the native defects in ZnO appear to migrate more easily under the electrical field, and voltage sweeps also cause unstable behavior in ZnO-based QLED devices.

Comment 2:

Based on previous research, it has been observed that the driven voltage of a device during aging experiences a rapid decrease in the initial stage, simultaneously with a substantial luminance increase. This process seems to be “quicker”. In light of this, I would like to suggest that the authors include a discussion of the voltage behavior during device aging, particularly in the initial stage, to further enhance the analysis presented in this manuscript. Since the degradation of Ag/ZnO interfaces seems only to increase the device resistance.

Comment 3:

In fact, there are numerous reports, including the authors' previous research, indicating that a fresh device with a ZnO electron transport layer can exhibit higher efficiency. However, based on the model proposed in this study, the occurrence of anomalous increase in luminance in ZnO-based QLEDs containing ZnO/Ag interface is difficult to avoid. Hence, it raises the question whether a fresh device with high efficiency, approaching the theoretical limit, would exhibit such anomalous increase in luminance. How the ligands migration effect device aging behavior in such device?

Comment 4:

The authors observe that electron injection becomes less efficient after the initial elevation in efficiency (Line 150). However, the increased efficiency under the same current stress implies that more carriers are being injected into the quantum dot (QD) to form exciton, since the efficiency elevation is not due to exciton quenching verified by in-situ/operando characterizations (Line 115). Further clarification on this contradiction is required from the authors.

Comment 5:

The degradation mechanisms appear to differ significantly between Red, Green, and Blue QLEDs, the author may wish to consider emphasizing the Red device in the title or main text for the purpose of clarity and focus.

Comment 6:

At pages 9-10, the authors suggest “The migration and diffusion of the excessive acetates during the device operation (bottom of Fig. 4b) would cause degradation of electrical contact, leading to the

gradual decrease of the electron-injection capability. " So What about other zinc oxides that don't use acetate as ligands? such as amines. There will still be failure and positive aging.

Comment 7:

Tof-sims can only obtain elemental information. Did the author draw red signals of acetate based on carbon? How to avoid interference from other carbons, such as surface contamination?

Comment 8:

Are fig4b, pristine and aged the same sample? If not, how do you distinguish between slight thickness differences between samples or interfacial reactions?

Comment 9:

What's the resolution of this SIMS ? what's the concentration of acetate in the ZnO film? Is this level detectable?

Response to Reviewer #1

In this work, the author claims that the anomalous efficiency elevation of QLEDs is induced by the degradation of the electron-transport layer/cathode interface. This conclusion is proved by using In-situ/operando characterizations, peel-off-and-rebuild experiments, and depth-profiling analyses. However, the author still needs to solve several key problems.

Our Response and Revision: We appreciate the reviewer for the enlightening and constructive comments, which greatly help to improve the quality of our manuscript. We would like to address the specific concerns in the following point-by-point responses.

Comment 1: *The efficiency elevation of QLEDs is a common phenomenon and has been observed in many reports. The author claims the anomalous efficiency elevation of QLEDs is induced by the interface of ZnO ETL and Ag electrode. However, the anomalous efficiency elevation of QLEDs was observed regardless of whether the ZnO electron transport layer was included in the device (ACS Nano, 2022, 16, 6, 9631-9639; Adv. Mater. 34, 4, 2106276). Therefore, it is unconvincing that the author think the anomalous efficiency elevation is completely induced by the interface of ZnO ETL and Ag electrode. The author must strictly prove whether the remaining functional layers and interfaces are unchanged, rather than just monitoring the PL of the QDs.*

Response 1: We concur with the reviewer's suggestions of assessing whether other functional layers and interfaces are unchanged. A series of new experiments were carried out to comprehensively probe the changes of the TFB, PEDOT:PSS and ZnO layers during efficiency elevation.

(1) Photoluminescence (PL) spectra of the TFB were performed to analyse the changes in the hole-transport layer (HTL) during ageing. A pristine QLED was subjected to the efficiency-elevation process, i.e., stressed at a constant current density of 50 mA cm^{-2} for 24 hours. PL spectra of the TFB layer show minimal changes after the efficiency-elevation process (from red curve to blue curve, Fig. R1). After further ageing of this device to a stage with pronounced luminance drop to 75% of the initial value (50 mA cm^{-2} for 300 hours), the PL spectrum of TFB shows (grey curve) discernible enhancements in the lower-energy emissions (arrows in Fig. R1). According to literature reports (*Synth. Met.* 2003, 139, 759–763, *J. Phys. Chem. B* 2004, 108, 25, 8689–8701, *Adv. Funct. Mater.* 2007, 17, 1, 71–78), such spectral features could result from electrochemical reduction reactions of polyfluorene polymers. These results indicate that while the degradation of TFB will occur after the long-term operation of our QLED, it does not contribute to the early stage of efficiency elevation in our device.

Fig. R1

(0) Raman analyses on the PEDOT:PSS hole-injection layer (HIL). Literature reports (*J. Phys. Chem. C* 2010, 114, 14, 6822–6830, *Phys. Chem. Chem. Phys.* 2022, 24, 541–550) suggest that chemical redox of PEDOT:PSS can be signified by shifts in the Raman spectra. During revision, we measured the Raman spectra of the PEDOT:PSS HILs in the pristine and aged QLEDs, respectively. The surfaces of PEDOT:PSS in the QLEDs were exposed by removing all the top layers, as illustrated in Fig. R2a. As shown in Fig. R2b, the PEDOT:PSS layers show similar Raman spectra in the pristine device (red curve) and after the efficiency-elevation process (blue curve). The results indicate no substantial

chemical changes in the PEDOT:PSS HILs.

Fig. R2

(3) X-ray photoelectron spectroscopy (XPS) analyses in the bulk of the ZnO ETLs. We conducted additional XPS depth profiling analyses on the pristine and aged QLEDs. The relative contents of depth profiling of Zn and Ag in pristine and aged devices are shown in Fig. R3a and d, respectively. We define the Ag/ZnO interface by the regions where the relative contents of Ag atoms drop from 90% to 10% (*Adv. Mater.* 2020, 2006178). This allows us to extract the XPS spectra that represent the properties in the bulk of the ZnO ETLs in both the aged sample and the pristine sample (the XPS scans corresponding to the solid grey circles in Fig. R3a and Fig. 3d). The Zn $2p$ spectra show identical peak positions (Zn $2p_3$ at 1021.5 eV and Zn $2p_5$ at 1044.6 eV) and the similar FWHMs (full width at half maximum, 1.5 eV) before (Fig. R3b) and after (Fig. R3e) the efficiency-elevation process. Moreover, according to previous reports on oxygen-related defects in ZnO (*J. Appl. Phys.* 2017, 121, 144503; *Appl. Phys. A* 2008, 90, 317–321; *Appl. Surf. Sci.* 2000, 158, 134–140), the O $1s$ peaks could be decomposed into three parts corresponding to the oxygen species on the surface of ZnO nanoparticles (O_{II} , 532.2 eV), O^{2-} ions near oxygen vacancies (O_{III} , 531.6 eV), and O^{2-} ions in the ZnO lattice (O_I , 530.2 eV), respectively. Gaussian fits on the O $1s$ spectra show similar compositions of the oxygen species with ~78% of O_I , ~16% of O_{II} , and ~6% of O_{III} in these devices, suggesting no significant changes in the chemical state of oxygen and oxygen-related defects.

Fig. R3

In the original manuscript, the *in-situ* EL-PL analyses indicate no significant enhancement in the PLQY of the QDs during the efficiency-elevation process (Fig. 2b). The peel-off-rebuild experiments show that by rebuilding the ZnO and Ag structures from an aged device, the device characteristics can recover from those of the aged status after efficiency elevation (Fig. 4b-e, revised manuscript). Together with the above supplementary analysis results, we conclude that the properties of HIL, HTL, QD emissive layers and ZnO ETLs remain largely unchanged after the efficiency-elevation process.

The above results are integrated as Supplementary Fig. 11–Supplementary Fig. 13 in the revised manuscript. We have also added one paragraph to discuss the absence of the changes in the other functional layers during the efficiency elevation (Line 266-277, main text). These additional results would help to reach a more rigorous conclusion of this work.

Comment 2: *The EQE is only 10% when the author studies the anomalous efficiency elevation of the device. However, the author has achieved EQE of 20% of the shelf-stable QLEDs in 2020 (Adv. Mater. 2020, 32, 52, 2006178). When the EQE exceeding 20%, the device does not show the anomalous*

efficiency elevation (Adv. Mater. 2020, 32, 52, 2006178, Figure 5e)?

Response 2: We appreciate this point about the comparison with our previous work. We would like to clarify that the efficiency-elevation phenomenon generally exists in our shelf-stable QLEDs, including the devices studied in this work (based on highly-conductive ZnO) and those reported in 2020 (*Adv. Mater. 2020, 32, 52, 2006178*, based on bilayer ZnO). In the *Adv. Mater.* paper, the devices were stressed under forward bias for several hours prior to the electrical measurements, which was described in the Characterization section (Supplementary Information, page 3). Essentially, the EQEs of 20% were achieved in those QLEDs that had already completed the efficiency-elevation process.

We also acknowledge that this comment raises an intriguing question, i.e., whether the efficiency-elevation phenomenon depends on the initial condition of the QLED. As unclosed in the discussion part, an essential reason that the deterioration of the ZnO/Ag interface can lead to an efficiency elevation in our shelf-stable QLED is that the pristine device is in a status with excessive distribution of electrons in the QDs. Such devices have a low initial exciton-generation efficiency ($\eta_x = N_x/N_e$, conversion ratio of electron into excitons), which can be increased upon the degradation of the electron-injection capability. As for an ideal device with near-unity internal quantum efficiency and η_x , the degradation of the electron-injection capability would not further improve the efficiency.

In the revised manuscript, we added one sentence to note that some high-efficiency QLEDs could result from pre-stressing by a short duration of electrical excitations (Line 62-65, main text). We also enriched the manuscript with additional discussions on the dependency of the efficiency-elevation phenomenon on the initial status of QLEDs (Line 320-335, main text).

Comment 3: *The in-situ/operando characterizations, peel-off-and-rebuild experiments have been reported many times (Nat. Photonics, 2022, 16, 505-511, Nat. Commun. 2020, 11, 2309, J. Phys. Chem.*

Lett. 2020, 11, 12, 4649–4654.) It will be necessary and important for the authors to clearly demonstrate the uniqueness and innovation compared to previous methods.

Response 3: We appreciate this constructive comment. We would like to highlight the uniqueness and innovation of our experimental techniques from three aspects.

First, the in-situ/operando characterizations in this work feature an “all-in-one” instrumentation (Fig. 2a) that enables multi-channel tracking of the changes in the PL intensity, EL intensity, EL spectra, and current-voltage (J-V) characteristics of an operating QLED in different ageing stages. This holistic method effectively mitigates device-to-device variations and provides abundant information for a comprehensive understanding of the evolving charge dynamics throughout the ageing process. In contrast, the in-situ characterizations in our previous reports served specific purposes. For example, the optical measurements reported in *Nat. Photon. 2022, 16, 505–511* were focused on the evolutions of the shape of the PL spectra, in which there was a lack of electrical characterizations during the long-term operation. Regarding the optical measurements in *Nat. Commun. 2020, 11, 2309*, the single-nanocrystal spectroscopy served specifically to probe the PL characteristics of EL devices at the single-QD level, which does not simultaneously monitor the various opto-electro properties of QLEDs as in this work.

Second, our peel-off-and-rebuild experiments manage to restore both the electrical and optical properties of a full stack of QLED. In the previous report (*J. Phys. Chem. Lett. 2020, 11, 12, 4649–4654.*), the peeled-off QLEDs were transformed into hole-only devices by depositing fresh electrodes, which verified the degradation of electrical properties of the TFB upon long-term ageing. In this work, by peeling off and rebuilding the full QLED, we can unambiguously pinpoint the degradation site responsible for both the elevated EL efficiency and impaired charge injection.

Third, during the revision, we introduced cross-sectional SKPM for *operando* characterizations of the

pristine and aged QLEDs (see Response 6, Page 10, for details). This method allows us to image the electric-potential and electric-field landscapes in active devices with high spatial resolution, complementary to the conventional optical and electrical measurements. To the best of our knowledge, this is the first report on the use of this powerful and unique tool in the QLED field.

Overall, this work combines a series of state-of-the-art *operando* and depth-profiling techniques to study the degradation mechanism, in which the non-invasive electrical and optical measurements are complementary to the invasive structural and chemical measurements. We believe that the experimental designs in this work could provide a general and powerful toolbox for studying degradation mechanisms in various LEDs.

In the revised manuscript, we have rewritten a few sentences in the introduction part to highlight the uniqueness of our characterization techniques (Line 81-86, main text). We also added one sentence to the summary part to highlight the implications of this work in the methodological aspect (Line 355356, main text).

***Comment 4:** Compared with a fresh device, a red-shifted EL spectrum was observed in aged devices. The author believes that this is due to the fraction of the electrical-generated X- is gradually decreased during the efficiency-elevation stage. This explanation is hard to understand. Even during the efficiency-elevation stage, the injection rate of electrons will decrease, but it should still be higher than the hole injection rate. Therefore, more and more electrons should be accumulated in QDs, which will lead to increased X⁻.*

Response 4: We thank the reviewer for raising this interesting question regarding the charge dynamics of QLED during efficiency elevation. In the original manuscript, our description might have caused confusion between the charge-injection rates and space-charge distributions. In short, during the

constant-current operation, the decreased electron-injection capability does not decrease the electron-injection rates, which is fixed by the constant total electron current from the cathode (J_e , Fig. R4a). Instead, the decreased electron-injection capability leads to a decrease in the space-charge concentrations of electrons in the QD layer. Below, we elaborate in detail how the decrease of electron-injection capability will alter the charge concentrations in the QD and eventually decrease the formation of X^- .

First, we acknowledge that the better electron injection/transport capabilities in the QLED would lead to a higher concentration of electrons in the QDs than that of holes. This discrepancy in the “accumulation rates” of electrons and holes only comes into play during the establishment of space-charge distributions in the QLED, which typically happens in the first few *microseconds* inside the device upon turn-on. When the device has reached electrostatic equilibrium, the better electron injection/transport capabilities would not further increase the electron concentrations in the QDs, because the total electron current (J_e) should always be identical to the total hole current (J_h) to comply with the continuity of the electrical circuit (Fig. R4a). In other words, the input of one electron in the cathode should always be accompanied by the extraction of one electron from the anode. Therefore, in a steady-state condition, the discrepancies in the injection/transport capabilities between electrons and holes will manifest as the discrepancies in their space-charge distributions but not as their injection rates.

In this work, we focus on the slow evolution (on a time scale of *hours*) of the steady-state conditions of the QLED in different stages of ageing, each stage of which is assumed to have reached electrical equilibrium. Therefore, the changes in the electron concentrations during long-term operations are not dominated by the absolute magnitude of *electron-injection capability* but result from the *changes in the electron-injection capability*. The degradation of the ZnO/Ag interface leads to a larger driving force required for electrons to cross the interface, i.e., an increased voltage drop at the electron-

injection interface. In compliance with the Poisson equation, more electrons would be accumulated in the electron-injection interface and fewer are distributed in the QDs in the steady state of the aged QLED.

Fig. R4

In the revised manuscript, we have modified Fig. 6 to highlight the different charge distributions between the pristine and aged devices. We have also rewritten the mechanism explanation part to clarify how the decreased electron-injection capability will lead to decreased electron concentrations in the QD layer at a constant injection rate (Line 280-306, main text).

Comment 5: Previous work believed that the leakage of electrons into TFB during operation will worsen the hole injection speed (*J. Phys. Chem. Lett.* 2020, 11, 12, 4649–4654.). Is the deterioration of hole injection rate slower than that of electron injection rate during efficiency-elevation stage?

Response 5: We appreciate this comment on the relative degradation rates of hole injection and electron injection. During revision, we have conducted additional experiments to probe the extent of degradation of TFB during the ageing process. As demonstrated in Fig. R1, while TFB can show

dramatic changes in the PL spectrum after long-term operations (300 hours, blue curve), we observed minimal changes in the PL of TFB during the efficiency-elevation process (24 hours, green curve). The results indicate that during the efficiency-elevation process, the TFB HTL used in this study is relatively stable and undergoes slower deterioration compared to that of the cathode/ETL interface. We would also like to note that the extent of electron leakage and the deterioration of TFB are highly dependent on the band structure of the QDs and the quality of the TFB materials. It is reasonable that our TFB materials are stable in the early stages of operation of the red QLEDs, which exhibit less electron leakage compared with green or blue QLEDs.

In the revised manuscript, we have provided the comparison of PL spectra of the TFB after different stages of ageing in Supplementary Fig. 11. We also added two sentences in the discussion part to highlight the different degradation rates of electron injection and hole injection in our QLEDs (Line 328-332, main text).

***Comment 6:** The author think that the incomplete recovery of the Ag-rebuilt device may be due to the residue of aged species at the ZnO/Ag interface. Is it possible that the electron injection speed was also reduced at the interface of ZnO and QDs? Due to the electron injection speed was reduced at the interface of ZnO and QDs, resulting in the EQE of aged Ag rebuilt device is higher that of the fresh Ag rebuilt.*

Response 6: We thank the reviewer for pointing out the possibility of deterioration of the QDs/ZnO interface, which cannot be completely ruled out by the experiments on the Ag-rebuilt devices. In response to this comment, we performed *operando* cross-sectional SKPM analyses on the QLEDs to obtain a complete picture of the changes in the electrical-potential landscapes during the efficiency-elevation process (Fig. R5a).

Fig. R5

The cross-sectional SKPM we employed (developed in Prof. Liwei Chen and Prof. Qi Chen's group) is a novel technique capable of scanning the electrical-potential distributions in operating optoelectronic devices with high spatial resolution (<30 nm). Briefly, the cross-sectional samples of the devices are delicately prepared by ion-beam milling, during which the device performance can be largely preserved. The surface potential of the cross-sectional sample is laterally scanned by using a conductive tip in the SKPM mode. In the meantime, a constant bias (V_{bias}) is applied between the cathode and anode of the device to enable *operando* measurements. This method has been successfully applied in various thin-film solar cells and photodetectors to image their electric-potential landscapes during operations (*Nat. Commun.* 2015, 6, 7745; *Nat. Commun.* 2019, 10, 4593; *Nano Lett.* 2021, 21, 19, 8474–8480; *Energy Environ. Sci.* 2022, 15, 2499–2507).

In our additional experiments, we prepared the cross-sectional sample of the pristine QLED and the aged QLED (after the efficiency-elevation process) via mechanical cleavage from the back of the

substrates, followed by argon-ion beam milling of the exposed edge in liquid nitrogen. According to the J-V measurements on the cross-sectional samples, the aged cross-sectional device shows a decreased slope in the J-V curve (Fig. R6a) and an increased ideality factor with respect to the pristine cross-sectional device (Fig. R6b), consistent with the trends observed in the efficiency elevation of the full-scale QLEDs (Fig. 2, main text). The results confirmed that the electrical properties of the devices are preserved in the cross-sectional samples.

Fig. R6

Fig. R5b-e show the height images and the corresponding phase images of the pristine and aged QLED, respectively. The high-quality QLED cross-sections with small fluctuations in the height channels and clear contrasts in the phase channels allow us to identify the different regions of electrodes and functional layers (interfaces indicated by dashed lines), which also enable reliable SKPM measurements on the surfaces. The surface potential images of the pristine and aged QLEDs under operational conditions (bias: 3 V) are shown by Fig. R5f and Fig. R5h, respectively. The partial derivatives of the surface potentials with respect to their lateral positions, which represent the electric fields normal to the charge-injection directions in the QLEDs, are extracted in Fig. R5g and Fig. R5i. Notably, a more rapid rise in the surface potential at the interface near Ag is observed in the operation of the aged device, which corresponds to an emerged strong electric field region near the ZnO/Ag interface in the lateral electric field images (indicated by the arrow in Fig. R5i). Meanwhile, there are only minor changes in the electrical potential and field distributions in the other functional layers. These features are more clearly seen in the representative profiles of the surface potential distributions (Fig. R5j) and the lateral electric field distributions (Fig. R5k). Meanwhile, there are only minor

changes in the electrical potential and field distributions in the other functional layers.

The above experiments reveal that the primary change in the electron-injection structure is the increased voltage drop at the ZnO/Ag interface, which also confirms the absence of degradation features at the ZnO/QD interface. The additional results are consistent with our depth-profiling experiments and are complementary to the peel-off-and-rebuild experiments.

In the revised manuscript, we have included the cross-sectional SKPM analyses as Fig. 3. We also added one section to describe the above new results (Line 172-200, main text).

***Comment 7:** The author believes that the reduced electron injection speed at the interface of ZnO and Ag is due to is due to the electrical contact degradation caused by the diffusion of Ac⁻ to the Ag electrode. Please provide some direct proof.*

Response 7: We acknowledge the query for direct proof of the degradation of the ZnO/Ag interface. We stress that it is challenging to directly observe the migration of organic species near an interface inside a multilayer device. In the original manuscript, we characterized the depth profiles of acetates by using ToF-SIMS, in which the aged QLED shows broadened distributions of acetates into the Ag electrode. To offer more direct and intuitive observations on this interface, we performed additional experiments to directly image the ZnO/Ag interface by using integrated differential phase contrast scanning transmission electron microscopy (iDPC-STEM).

The conventional high-angle annular dark-field (HAADF) STEM techniques primarily detect the signals from heavy atoms. It is thus challenging to utilize HAADF-STEM to image the interface with organic species consisting of light atoms. By contrast, iDPC-STEM is an advanced phase-imaging technique based on STEM that enables simultaneous imaging of both light and heavy atoms. This is

because iDPC-STEM directly measures the phase shift of the transmitted electron wave caused by the sample, which is linear to the local electrostatic potential field of a thin sample (*Ultramicroscopy* 2016, 60, 265-280). In previous reports, iDPC-STEM has been a powerful imaging tool for samples with visualization of both heavy and light atoms (*Adv. Funct. Mater.* 2019, 29, 1903843; *Nature* 2021, 592, 541–544; *J. Struc. Bio.* 2022, 24, 07837), and is suitable for the ZnO/Acetate/Ag interface in our case.

Fig. R7a shows the HAADF-STEM cross-sectional image of a QLED, in which the contrast at the ZnO/Ag interface is dominated by the difference in atomic numbers. The iDPC-STEM images successfully unveiled the morphologies of the ZnO/Ag interfaces before (Fig. R7b) and after the efficiency-elevation process (Fig. R7c). Notably, compared with the relatively smooth and flat interfacial region in the pristine device, a “trench” (manifests as an undulation of contrast) appears on the Ag side of the ZnO/Ag interface. These observations confirm the deterioration of the ZnO/Ag interface, supporting the depth-profiling results of acetate diffusion into the Ag.

Fig. R7

In the revised manuscript, we have added the iDPC-STEM images into Fig. 5a to provide a first glance at the deterioration of the interface. We also added few sentences (Line 224-233, main text) to describe the direct observations on the ZnO/Ag interface.

Comment 8: Please check the manuscript carefully. The $C2H3O2''+$ in the Figure 4b should be changed to $C2H3O2''-$.

Response 8: We thank the reviewer for pointing out this typo. Our ToF-SIMS measurements were conducted in the positive ion mode with “**MC⁺** Approach”. It detects cluster ions that are composed of the sample segment of **M** in combination with the primary ions of **Cs⁺** or **Cs₂⁺**. The signals from the cluster ions of **Cs₂C₂H₃O₂⁺** are used to probe the acetate distributions in the devices.

In the revised manuscript, we have changed the notations to more explicit expressions, **[M+Cs]⁺** or **[M+Cs₂]⁺**, for the species detected in the ToF-SIMS. Similar notations are used in previous reports of ToF-SIMS (*Nature* 2019, 571, 245–250; *Nat. Rev. Chem.* 2020, 4, 257–268).

Response to Reviewer #2

This manuscript presents a detailed analysis of the microscale physical processes responsible for the anomalous increase in luminance observed during the aging of quantum dot light-emitting diodes (QLEDs). The authors conclude that the primary cause of this phenomenon is the migration of ligands in the metal oxide nanocrystal layer, which leads to degradation at the ZnO/Ag interface. The presented argument is supported by a series of verification experiments, leading to a logical and self-consistent narrative.

It is important to note, however, that the mechanisms underlying device degradation are complex, and the model proposed in this study may not fully capture all relevant factors. Compared to the experiment skills and characterization, the discussion of this paper is weak.

To ensure the manuscript's high-quality and enhance its impact, the reviewer suggests that authors address the following questions and provide a more comprehensive discussion, then resubmitted.

Our Response and Revision: We thank the reviewer for acknowledging our verification experiments and characterizations of this anomalous phenomenon. We agree with the reviewer's assessment that a more comprehensive discussion shall improve the quality and enhance the impact of our manuscript. During revision, we performed additional experiments to address all the reviewer's comments, which eventually helped us to enrich the discussion part. Point-by-point responses are provided below.

Comment 1: *As discussed in the introduction section, similar phenomena have been widely reported in previous literature. Under current stress, the efficiency of quantum dot light-emitting diodes (QLED) is promoted. Some previous reports have discussed the behavior and attributed it to electrical unstable*

oxygen-related defects in ZnO-based nanocrystals. In this study, the electrical aging behavior of QLED is analyzed, and these factors cannot be overlooked. Specifically, the native defects in ZnO appear to migrate more easily under the electrical field, and voltage sweeps also cause unstable behavior in ZnO-based QLED devices.

Response 9: We agree with the reviewer that the efficiency elevation of QLED has been reported in previous literature. We would like to highlight that the efficiency elevation reported in this work is observed on shelf-stable QLEDs and is induced solely by electrical stressing. This is distinct from the “positive ageing” effects induced by migration of oxygen-related defects as mentioned in some previous reports, where the QLED can show efficiency enhancement during storage without electrical excitation (e.g., *Appl. Phys. Lett.* 2020, 117, 093501).

We also appreciate the reviewer’s suggestion to investigate the possibility that the oxygen-related defects in the ZnO nanocrystals might have an impact on the efficiency elevation in our device. The electrical migration of oxygen-related defects, e.g., oxygen vacancies, was observed in previous reports on ZnO nanocrystals (*Appl. Phys. Lett.*, 2006, 89, 102103). Since the concentration of oxygen vacancies influences the conductivity of ZnO, a typical manifestation of the oxygen-vacancy migration is the “resistive switching characteristics”, i.e., the devices show hysteresis in the J-V curves acquired in forward and reverse sweeps (*Phys. Status Solidi A*, 2010, 207, 484-487, *J. Appl. Phys.* 2017, 121, 144503, *Appl. Phys. Lett.* 2020, 117, 093501).

During revision, we performed the forward and reverse scans of the J-V curves of the QLEDs. As shown below, the J-V curves obtained from the forward scans (blue curves) and reverse scans (red curves) are nearly identical for both the pristine device (Fig. R8a) and the aged device (Fig. R8b, after the efficiency elevation). The absence of resistive switching effects suggests that migration of oxygen-

related defects is not a prominent factor in our QLEDs.

Fig. R8

Additionally, we analysed the changes of oxygen-related defects in the ZnO ETLs before and after the efficiency-elevation process by using depth-profiling XPS on the QLEDs. According to previous reports on oxygen-related defects in ZnO (*J. Appl. Phys.* 2017, 121, 144503; *Appl. Phys. A* 2008, 90, 317–321; *Appl. Surf. Sci.* 2000, 158, 134–140), the O 1s peaks could be decomposed into three parts corresponding to the oxygen species on the surface of ZnO nanoparticles (532.2 eV), O²⁻ ions near oxygen vacancies (531.6 eV), and O²⁻ ions in the ZnO lattice (530.2 eV), respectively. Fig. R9a and Fig. R9d show the relative contents of Zn and Ag in pristine and aged devices, respectively, obtained from depth-profiling XPS. We define the Ag/ZnO interface by the regions where the relative contents of Ag atoms drop from 90% to 10% (*Adv. Mater.* 2020, 2006178). This allows us to extract the XPS spectra that represent the properties in the bulk of the ZnO ETLs in both the aged sample and the pristine sample (the XPS scans corresponding to the solid grey circles in Fig. R9a and Fig. R9d). The Zn 2p spectra before (Fig. R9b) and after (Fig. R9e) the efficiency-elevation process shows identical peak positions and peak widths. Gaussian fits of the O1s spectra show similar compositions before (Fig. R9c) and after (Fig. R9f) efficiency elevation, with the proportions of the oxygen vacancies being ~6% in both devices. These results suggest no significant changes in the chemical state and

concentration of the oxygen-related defects inside the ZnO ETL after the efficiency-evaluation process.

Fig. R9

Based on the above results, we conclude that the oxygen-related defects in the ZnO ETLs are not likely to be the primary cause of the observed efficiency elevation in our shelf-stable QLEDs. The absence of the migration of oxygen-related defects in this work might be due to that the surface defects on our ZnO NPs are better passivated. Moreover, the ZnO ETLs in this study are deposited in a nitrogen atmosphere and the QLEDs are encapsulated before the test to avoid exposure to ambient oxygen. It has been demonstrated in the literature that the ZnO-based devices could show much reduced J-V hysteresis in the absence of ambient oxygen compared with that in air (*J. Appl. Phys.* 2014, 116, 114501).

In the revised manuscript, we have included the above results in Supplementary Fig. 14 and Supplementary Fig. 13. We also added two sentences to note the minimal changes in the oxygen-related defects in the ZnO and the absence of resistive switching effects (Line 270-275, main text).

Comment 2: *Based on previous research, it has been observed that the driven voltage of a device during aging experiences a rapid decrease in the initial stage, simultaneously with a substantial luminance increase. This process seems to be “quicker”. In light of this, I would like to suggest that the authors include a discussion of the voltage behaviour during device aging, particularly in the initial stage, to further enhance the analysis presented in this manuscript. Since the degradation of Ag/ZnO interfaces seems only to increase the device resistance.*

Response 10: We appreciate the valuable suggestion to include a discussion on the voltage behaviour during the initial stage of device ageing, an aspect that enhances the comprehensiveness of our analysis.

To address this point, we extracted the voltage behaviour of our shelf-stable QLEDs during the initial stage of device ageing. As depicted in Fig. R10a, the device shows a rapid decrease in the driving voltage accompanied by a decrease in luminance during this initial stage. Subsequently, both the driving voltage and luminance gradually increase in the efficiency-elevation process. The “quicker” process of voltage drop is in line with previous reports (e.g., *ACS Nano* 2018, 12, 10, 10231–10239; *J. Mater. Chem. C* 2020, 8, 2014; *J. Chem. Phys.* 2023, 158, 131101). According to the literature, the quick process at the initial stage of QLED operation can be attributed to the rapid negative charging of QDs, which promotes nonradiative Auger recombination (resulting in a rapid drop of luminance), Coulombic-interaction enhanced hole injection (resulting in a quick decrease of driving voltage), or temperature elevation caused by Joule heating (resulting in loss of the PL QY of QDs).

We note that the “quicker” process at the initial stage of the ageing of our QLED is largely reversible, and it shall not contribute to the long-term operational instability of the device. Fig. R10b shows the EL intensity evolutions of the QLED under pulsed electrical stressing, alternating between 20 min of constant current (100 mA cm⁻²) and 2 min of relaxation (grey-shaded regions, bias: 0 V). The rapid decay in EL intensity repeatedly occurs in the early stage of turn-on after every relaxation period, while

the device shows increased EL intensity in a longer time scale. These features make the quick process substantially different from the efficiency-elevation process which is irreversible and could last for hundreds of hours. Therefore, we suggest that the mechanism associated with the quick process during the initial stage of device ageing and the mechanism associated with the efficiency-elevation process should be investigated separately. In this work, we would like to focus on the long-term efficiency-elevation processes and identify its main cause as the degradation of electron-injection capability at the electron-transport layer/cathode interface.

Fig. R10

In the revised manuscript, we have added a few sentences to note the voltage behaviour of the and the difference between the long-term efficiency-elevation process and the temporal behaviour of the QLED (Line 163-170, main text). The above results are provided as Supplementary Fig. 4.

Comment 3: *In fact, there are numerous reports, including the authors' previous research, indicating that a fresh device with a ZnO electron transport layer can exhibit higher efficiency. However, based on the model proposed in this study, the occurrence of anomalous increase in luminance in ZnO-based QLEDs containing ZnO/Ag interface is difficult to avoid. Hence, it raises the question whether a fresh device with high efficiency, approaching the theoretical limit, would exhibit such anomalous increase in luminance. How the ligands migration effect device aging behaviour in such device?*

Response 11: We thank the reviewer for raising this interesting question. In our experience, fresh

devices with ZnO ETLs exhibiting high efficiency often result from the so-called “positive ageing” effects, which effectively improve device performance during storage. Our in-depth mechanism study (*Adv. Mater.* 2020, 32, 2006178) has unveiled that the “positive ageing” effects involve in-situ chemical reactions induced by the acids contained in encapsulation resin. Consequently, multiple processes can take place in these devices, leading to complex and often storage-time-dependent device-ageing behaviours. We highlight that the shelf-stable red QLEDs (*Adv. Mater.* 2020, 32, 2006178 and this study) offer an ideal platform for investigating the mechanisms that govern the operation-induced changes, including the anomalous efficiency-elevation processes. The high-efficiency shelf-stable QLEDs reported in *Adv. Mater.* 2020, 32, 2006178 had already completed the efficiency-elevation process (see Response 2 for details).

We would also like to note that QLEDs can exhibit no efficiency-elevation process despite of the degradation of the ZnO/Ag interface. As unclosed in the discussion part (Fig. 6), an essential reason that the deterioration of the ZnO/Ag interface can lead to an efficiency elevation in our shelf-stable QLED is that the pristine device is in a status with excessive distribution of electrons in the QDs. Such a device would undergo increased exciton-generation efficiency ($\eta_X = J_{e-X}/J_e$, conversion ratio of electron currents into exciton-recombination currents) upon degradation in the electron-injection capability (see Response 4 for a more detailed explanation). Regarding a hypothetical fresh device with an efficiency approaching the theoretical limit, it would be reasonable to assume that the exciton-generation efficiency has already approached near-unity ($J_{e-X} \approx J_e$). For such a device, we would expect the occurrence of ligand migration at the ZnO/cathode interface, but the degradation of the electron-injection capability would unlikely further improve the η_X and the EQE.

In the revised manuscript, we have added additional discussion about the occurrence of efficiency elevation in QLEDs (Line 320-335, main text). We have also noted that some high-efficiency QLEDs rely on a pre-stressing process by electrical excitation to reach an optimal operation condition (Line 62-65, main text).

Comment 4: *The authors observe that electron injection becomes less efficient after the initial elevation in efficiency (Line 150). However, the increased efficiency under the same current stress implies that more carriers are being injected into the quantum dot (QD) to form exciton, since the efficiency elevation is not due to exciton quenching verified by in-situ/operando characterizations (Line 115). Further clarification on this contradiction is required from the authors.*

Response 12: We thank the reviewer for pointing out the confusing description of the mechanism, especially in terms of “efficiency” and “efficient”. To clarify this, we would like to elaborate on how the space-charge distributions of electrons (in different functional layers) would affect the conversion ratio of electron currents into excitons. The efficiency-elevation process of our QLED is a slow transition of the operation conditions from the pristine quasi-steady state to the aged quasi-steady state, in which both the space-charge distributions and the conversion of electron currents are changed.

In the original manuscript (line 150), the statement “electron injection is less efficient” was intended to convey that a higher voltage is required to drive the QLED at the same total current density after the efficiency-elevation process because a larger driving force is required for electrons to overcome the electron-injection barrier. The voltage drop at the interface is connected to the space-charge densities through the Poisson equation. This means that more electrons are distributed at the electron-injection interface and fewer electrons are distributed in the QDs layer after the efficiency-elevation process (Fig. R11a to Fig. R11c), while the total electron-injection rate (J_e) is fixed by the constant-current driving mode. Therefore, we described the electron-injection capability as “less efficient” essentially to capture the reduced concentration of electrons in the QD layer, instead of the reduced electron-injection rates. The decreased space-charge distribution of electrons in the QDs leads to a lowered probability of forming X^- in the QDs (decreased non-radiative recombination currents via Auger recombination, J_{e-}

Auger) and a lowered probability of electron transfer from QDs into TFB (decreased leakage current out of the QD layer, $J_{e\text{-Leakage}}$), both of which are dependent on the concentration of electrons in the QD layer. Consequently, due to the suppression of these two efficiency-loss channels, the conversion ratio from the electron currents (J_e) to exciton-recombination currents (J_{e-x}), i.e., exciton-generation efficiency (η_x), is increased. Given the constant J_e input from the cathode, a higher η_x would lead to more excitons being generated per unit of time, and thus an increased luminance.

Fig. R11

We have revised Fig. 6 to illustrate the different dissipation channels (J_{e-x} , $J_{e\text{-Leakage}}$ and $J_{e\text{-Auger}}$) of the total electron currents (J_{e-x}). We have also rewritten the mechanism explanation part to clarify how the exciton-generation efficiency is increased during constant-current operation (Line 280-306, main text).

Comment 5: *The degradation mechanisms appear to differ significantly between Red, Green, and Blue QLEDs, the author may wish to consider emphasizing the Red device in the title or main text for the purpose of clarity and focus.*

Response 13: We agree with the reviewer that the degradation mechanisms can be different between red, green and blue QLEDs. Following the reviewer's suggestion, we have clarified in the abstract and the main text (highlighted, main text) of the revised manuscript to underscore that this study focuses on the efficiency-elevation of red QLEDs.

Comment 6: *At pages 9-10, the authors suggest "The migration and diffusion of the excessive acetates during the device operation (bottom of Fig. 4b) would cause degradation of electrical contact, leading to the gradual decrease of the electron-injection capability. " So what about other zinc oxides that don't use acetate as ligands? such as amines. There will still be failure and positive aging.*

[Redacted]

[Redacted]

[Redacted]

***Comment 7:** ToF-sims can only obtain elemental information. Did the author draw red signals of acetate based on carbon? How to avoid interference from other carbons, such as surface contamination?*

Response 15: We thank the reviewer for raising the questions concerning our ToF-SIMS measurements and the analyses of signals of acetate segments. ToF-SIMS detects the secondary particles from the outermost surface of the sample according to their mass-to-charge ratios (m/z),

which can probe the depth distributions of both ions and organic fragments with known m/z (*Adv. Mater.* 2014, 26, 5155–5159, *Nat. Energy* 2016, 1, 16118, *Nature* 2018, 562, 245–248, *Nat. Commun.* 2021, 12, 4868). In this work, an advanced detection mode based on the “ \mathbf{MCs}^+ Approach” was employed in the ToF-SIMS. Briefly, the cluster ions composed of the primary ions (\mathbf{Cs}^+) and the measured species (\mathbf{M}), i.e., \mathbf{MCs}^+ or \mathbf{MCs}_2^+ ions, were monitored. This approach can mitigate the matrix effect and improve the accuracy of depth profiles across interfaces (*J. Appl. Phys.* 1988, 64, 3760-3762; *Int. J. Mass. Spectrom.* 1995, 143, 11-18; *Appl. Surf. Sci.* 2008, 255, 1412-1414).

In Fig. 5c, the acetate signals were extracted from the mass-spectrum peaks corresponding to the m/z ratio of 324.84 (mass spectrum shown in Fig. R14), which agrees precisely with the \mathbf{MCs}_2^+ cluster ions of $[\text{C}_2\text{H}_3\text{O}_2+\text{Cs}_2]^+$ (relative atomic mass: 324.84617). Therefore, the acetate signals represent the distribution of acetates with no interference from other carbon sources.

Fig. R14

In the revised manuscript, we added one sentence to explain the detection mode of the ToF-SIMS measurements (Line 237-239, main text). We mentioned the specific m/z ratios corresponding to the segments in the Methods section (Line 442-444, main text). We also revised the expressions of the chemical species in the ToF-SIMS results (in the form of $[\mathbf{M}+\mathbf{Cs}_n]^+$) to enable clearer notations of the detected cluster ions.

***Comment 8:** Are fig4b, pristine and aged the same sample? If not, how do you distinguish between slight thickness differences between samples or interfacial reactions?*

Response 16: The pristine and aged samples shown in Fig. 4b were measured from two devices from the same batch. We agree with the reviewer that it is necessary to determine whether the depth-profile differences originated from thickness variations of the two samples. In response to this point, we repeated the ToF-SIMS measurements on multiple QLEDs. It is consistently observed that the aged device shows extended distributions of acetates compared with the pristine device (Fig. R15).

Fig. R15

Moreover, we also analyzed the depth profiles of a QLED which was subjected to different ageing durations (current density of 50 mA cm⁻²). As shown in Fig. R16, the depth profiles show a pronounced trend of acetate migration towards Ag as the ageing time increases from 0 h to 100 h. These additional measurements indicate the changes in acetate distributions resulted from the ageing processes but not the thickness variations of the samples.

Fig. R16

The above new results have been included as Supplementary Fig. 9 in the revised manuscript. We also added one sentence (Line 249-251, main text) to mention the trend of acetate migration as the ageing time prolongs.

Comment 9: *What's the resolution of this SIMS? what's the concentration of acetate in the ZnO film?*

Is this level detectable?

Response 17: We appreciate the reviewer's inquiry on the resolution of our ToF-SIMS measurements. The mass analyzer used in our ToF-SIMS characterizations provides a mass resolution of $m/\Delta m > 15,000$. The depth resolution of our ToF-SIMS measurements is < 1 nm according to the slow and steady sputtering rate of $\sim 1 \mu\text{m h}^{-1}$ and the data-acquisition rate of > 30 scans min^{-1} . The limit of detection of the ToF-SIMS equipment is on the order of 10^{-9} – 10^{-6} (ppb to ppm) in terms of concentration in the thin film (<https://www.iontof.com/tof-sims-secondary-ion-mass-spectrometry-leis-technique-low-energy-ion-scattering.html>). Therefore, the ToF-SIMS measurements possess sufficient spatial and concentration resolutions to probe the depth profiles of the species at the ZnO/Ag

interface. Meanwhile, we note that without standard control samples, ToF-SIMS cannot determine the absolute quantities of the chemicals.

We also thank the reviewer's suggestion of checking whether the concentration of acetates in the ZnO is detectable. In response to this, we conducted FTIR measurements for a quantitative analysis of the absolute concentration of acetate within the ZnO film. Specifically, the absorbance spectrum (Fig. R17b) of a ZnO film (~60 nm in thickness) was collected in the reflection mode (Fig. R17a), where the incident angle of the infrared light is 80° and the equivalent light path (b) across the ZnO film is approximated as 6.9×10^{-5} cm. The measured absorbance (A) corresponding to the C=O stretching vibration (~ 1600 cm^{-1}) in the ZnO film is 0.056 (Fig. R17b). The concentration of acetate (c) is then determined to be $\sim 1.8 \times 10^{-3}$ mol cm^{-3} by solving the Lambert-Beer equation ($A=\epsilon bc$), in which the molar absorption coefficient of acetate (ϵ) is taken from a previous literature (4.5×10^2 $\text{M}^{-1} \text{cm}^{-1}$, *Anal. Chim. Acta* 1993, 280, 253-261). We believe that the concentration of acetate in our sample is well above the limit of detection of the ToF-SIMS measurement.

Fig. R17

In the revised manuscript, we have enriched the Methods part with more detailed information on the resolution, detection limit, and sputter rate of the ToF-SIMS measurements (Line 438-442, main text). We also added two sentences to note the estimated acetate concentration (Line 247-249 main text).

REVIEWERS' COMMENTS

Reviewer #1 (Remarks to the Author):

I have read the revision and response carefully, this manuscript was revised well and replied my concerns. Thus, this work can be accepted to publish.

Reviewer #2 (Remarks to the Author):

The reviewer has carefully examined the authors' response to the initial comments and the revisions made to the manuscript. It is acknowledged that the authors have made a substantial effort to clarify and strengthen their work. The fundamental issues identified in the original manuscript have been addressed in the revised version. The manuscript now presents a more comprehensive and self-consistent analysis of the physical processes driving the observed phenomena in red quantum dot light-emitting diodes (QLEDs).

The revisions and the addition of Supplementary Information have improved the paper's quality and clarity. The enhanced discussions, particularly concerning the complexities of device degradation, provide context to the research findings.

The reviewer recommends accepting the manuscript for publication after a thorough language check.

Response to Reviewer #1

I have read the revision and response carefully, this manuscript was revised well and replied my concerns. Thus, this work can be accepted to publish.

Our response: We appreciate the reviewer again for the positive comment. No modification is made.

Response to Reviewer #2

The reviewer has carefully examined the authors' response to the initial comments and the revisions made to the manuscript. It is acknowledged that the authors have made a substantial effort to clarify and strengthen their work. The fundamental issues identified in the original manuscript have been addressed in the revised version. The manuscript now presents a more comprehensive and self-consistent analysis of the physical processes driving the observed phenomena in red quantum dot light-emitting diodes (QLEDs).

The revisions and the addition of Supplementary Information have improved the paper's quality and clarity. The enhanced discussions, particularly concerning the complexities of device degradation, provide context to the research findings.

The reviewer recommends accepting the manuscript for publication after a thorough language check.

Our response: We thank the reviewer again for recognizing our efforts on the revisions.

Following the suggestion, we have performed a thorough language check with a native speaker.